# TICKING ALL THE BOXES: GENERATED CHECKLISTS IMPROVE LLM EVALUATION AND GENERATION

## ABSTRACT

Given the widespread adoption and usage of Large Language Models (LLMs), it is crucial to have flexible and interpretable evaluations of their instruction-following ability. Preference judgments between model outputs have become the de facto evaluation standard, despite distilling complex, multi-faceted preferences into a single ranking. Furthermore, as human annotation is slow and costly, LLMs are increasingly used to make these judgments, at the expense of reliability and interpretability. In this work, we propose **TICK** (**T**argeted **I**nstruct-evaluation with **ChecK**lists), a *fully automated, interpretable* evaluation protocol that structures evaluations with LLM-generated, instruction-specific checklists. We first show that, given an instruction, LLMs can reliably produce high-quality, tailored evaluation checklists that decompose the instruction into a series of YES/NO questions. Each question asks whether a candidate response meets a specific requirement of the instruction. We demonstrate that using TICK leads to a significant increase (46.4% → 52.2%) in the frequency of exact agreements between LLM judgements and human preferences, as compared to having an LLM directly score an output. We then show that **STICK** (**S**elf-**TICK**) can be used to improve generation quality across multiple benchmarks via self-refinement and Best-of-N selection. STICK self-refinement on LiveBench reasoning tasks leads to an absolute gain of +7.8%, whilst Best-of-N selection with STICK attains +6.3% absolute improvement on the real-world instruction dataset, WildBench. In light of this, structured, multi-faceted self-improvement is shown to be a promising way to further advance LLM capabilities. Finally, by providing LLM-generated checklists to human evaluators tasked with directly scoring LLM responses to WildBench instructions, we notably increase inter-annotator agreement (0.194 → 0.256).

## 1 INTRODUCTION

Instruction-tuned Large Language Models (LLMs) are widely used as conversational assistants, where users expect responses to closely follow their intents (Wei et al., 2022a; Mishra et al., 2022; Bai et al., 2022a; Ouyang et al., 2022). The broad usage of LLMs creates a critical demand for reliable, flexible, and transparent ways of evaluating their instruction-following abilities. However, standard evaluation methods, such as preference labeling (Ouyang et al., 2022), direct scoring (Novikova et al., 2018; Wang et al., 2023b), and Elo rating (Bai et al., 2022a; Glaese et al., 2022), tend to obscure the reasoning behind evaluations. These methods also often result in significant disagreements, both among human annotators (Hosking et al., 2024) and between models and humans (Qin et al., 2024; Zheng et al., 2023).

To address these limitations, we introduce **TICK** (**T**argeted **I**nstruct-evaluation with **ChecK**lists), a novel approach to LLM-as-judge evaluation that uses the judge LLM to decompose instructions into checklists consisting of a series of YES/NO evaluation questions. These checklists provide *interpretable, fine-grained* assessments of whether a model response satisfies specific requirements of the instruction. Crucially, TICK eliminates manual effort in checklist creation, a substantial cost for existing checklist-based benchmarks (Qin et al., 2024; Wen et al., 2024). We rigorously demonstrate that current LLMs can already generate checklists matching the quality of human-written ones. In experiments, we show that using TICK leads to an absolute increase in the frequency of exact agreements between an LLM judge and human preferences of 5.8%.

For each instruction:

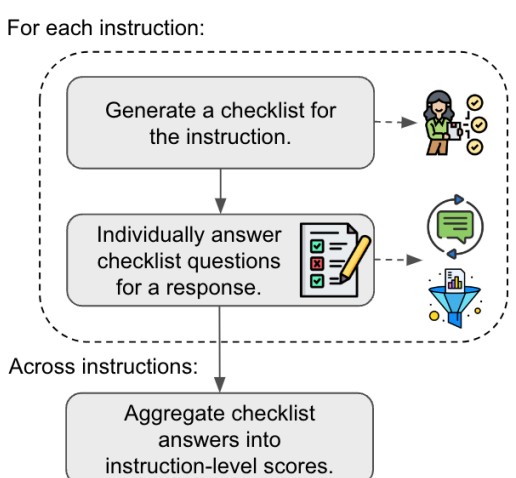

Generate a checklist for the instruction.

Individually answer checklist questions for a response.

Across instructions:

Aggregate checklist answers into instruction-level scores.

**Instruction**

Summarize the first season of the TV Series "Mr. Robot" using a similar tone to the show's main character, Elliot Alderson. Use subheadings and bullet points for formatting, and stay under 500 words.

**GPT-4o Generated Checklist**

- Is the response a summary of the first season of the TV series "Mr. Robot"?
- Is the response written in a tone similar to the show's main character, Elliot Alderson?
- Does the response use subheadings and bullet points for formatting?
- Is the response under 500 words?

Figure 1: **Left:** Diagram of TICK and its downstream uses of augmenting human evaluation, performing self-refinement, and response filtering. **Right:** Example of a generated evaluation checklist.

Building on this, we introduce **STICK** (**S**elf-**TICK**), an approach to in-context self-improvement where LLMs iteratively refine their responses based on TICK self-evaluations. We demonstrate that STICK enables LLMs to achieve significant performance gains across several benchmarks without the need for dataset-specific prompting or pre-existing human-written checklists. Specifically, Command-R+ shows a 6.5% absolute improvement on InFoBench (Qin et al., 2024) and a 7.1% absolute gain on WildBench (Lin et al., 2024), outperforming vanilla Self-Refine (Madaan et al., 2023). On LiveBench (White et al., 2024), STICK refinements enable Command-R+ to achieve a 3.8% improvement and GPT-4o to gain 0.8%, whereas vanilla Self-Refine leads to substantial degradation. These improvements span tasks for which in-context self-improvement has previously proven challenging, such as mathematics, reasoning, and coding (Huang et al., 2024; Kamoi et al., 2024; Tyen et al., 2024), as well as amplifying improvements on tasks that have previously been shown to benefit from self-critiques, such as constrained instruction-following (Madaan et al., 2023). When used for Best-of-N response self-selection, STICK improves on greedy decoding by 5.1% on InFoBench and 5.3% on WildBench, and even outperforms selection by a general-purpose reward model. Finally, we explore whether LLM-generated checklists can assist human evaluators by augmenting the annotation process and find significant improvements to inter-annotator agreement.

In summary, we make the following contributions:

1. We rigorously show that LLMs can generate evaluation checklists similar in quality to those written by trained human annotators across multiple diverse instruction-following datasets.

2. We introduce TICK, a checklist-based, automatic evaluation protocol that yields stronger agreement with humans than other general-purpose LLM-as-judge evaluations. Because TICK can be fully automated, it is cheaper and faster to run than existing checklist-based evaluations, and can be applied to arbitrary instruction-following datasets.

3. We leverage Self-TICK (STICK) to substantially improve instruction-following ability via self-refinement and Best-of-N selection on multiple challenging benchmarks.

4. We explore using LLM-generated checklists to assist human evaluators tasked with directly scoring an LLM output, and find that this improves inter-annotator agreement.

## 2 RELATED WORK

**Instruction-Following Evaluation:** There have been many efforts to improve the evaluation and benchmarking of LLMs' instruction-following ability. Some of these benchmarks aggregate instruction sets from a diverse range of sources to measure general instruction-following ability and use a judge LLM to score outputs (Li et al., 2023; Chia et al., 2023; Lin et al., 2024). Others decompose instructions into checklists, made up of YES/NO questions or PASS/FAIL criteria that a response

| Tasks | Command-R+ | | | GPT-4o | | |
|---|---|---|---|---|---|---|
| | Base | Self-Refine | STICK | Base | Self-Refine | STICK |
| Overall | 32.0 | 23.7 ($\downarrow$ 8.3) | **35.8** ($\uparrow$ 3.8) | 55.4 | 47.1 ($\downarrow$ 8.3) | **56.2** ($\uparrow$ 0.8) |
| Coding | 18.8 | 9.1 ($\downarrow$ 9.7) | **22.7** ($\uparrow$ 3.9) | 50.4 | 36.4 ($\downarrow$ 14.0) | **51.6** ($\uparrow$ 1.2) |
| Data Analysis | 25.9 | 5.3 ($\downarrow$ 20.6) | **29.8** ($\uparrow$ 3.9) | 52.4 | 27.2 ($\downarrow$ 25.2) | **52.5** ($\uparrow$ 0.1) |
| Instructions | 69.6 | 60.5 ($\downarrow$ 9.1) | **75.8** ($\uparrow$ 6.2) | 73.3 | 62.8 ($\downarrow$ 10.5) | **76.2** ($\uparrow$ 2.9) |
| Language | **24.6** | 13.8 ($\downarrow$ 9.8) | 24.1 ($\downarrow$ 0.5) | 50.9 | **51.4** ($\uparrow$ 0.5) | 50.4 ($\downarrow$ 0.5) |
| Mathematics | 23.7 | 23.6 ($\downarrow$ 0.1) | **25.5** ($\uparrow$ 1.8) | 52.3 | 51.8 ($\downarrow$ 0.5) | **53.1** ($\uparrow$ 0.8) |
| Reasoning | 29.2 | 30.0 ($\uparrow$ 0.8) | **37.0** ($\uparrow$ 7.8) | **53.3** | 52.7 ($\downarrow$ 0.6) | 53.3 (0) |

Table 1: A single step of self-refinement on LiveBench with Command-R+ and GPT-4o, using STICK to form self-critiques. Unstructured self-critiques are included as a baseline (Self-Refine), along with each LLM's base performance.

should meet (Zhou et al., 2023; Jiang et al., 2024; Qin et al., 2024; Wen et al., 2024). For example, the WildBench (Lin et al., 2024) dataset pairs its instructions with checklists (generated by two LLMs, then reviewed by humans) that are included in the evaluation prompt to get a score or preference from a judge LLM, but not explicitly answered or used to form a metric. Approaches to evaluation used in these works are therefore hard to make use of outside of the benchmarks themselves, relying heavily on humans for instruction and checklist curation. We instead design a dynamic evaluation protocol that can be employed on-the-fly and therefore seamlessly integrated into custom evaluation workflows, or used to steer high quality generation. Meanwhile, we avoid the cost of having humans in the loop and enable our evaluation quality to improve as LLMs improve.

**Language Critiques:** LLM critiques are an intuitive way of addressing the "black box" nature of evaluations. These critiques are intended to point out the strengths and weaknesses of outputs generated by the same LLM, or a different LLM. Critiques can be used to improve the quality of overall evaluations performed by an LLM judge or reward model (Ankner et al., 2024; Bai et al., 2022b; Wang et al., 2023a; Ye et al., 2024; Sun et al., 2024), inform human evaluations (Saunders et al., 2022; McAleese et al., 2024), or provide feedback that can be used to refine a response in-context (Scheurer et al., 2023; Tian et al., 2024; Madaan et al., 2023; Yuan et al., 2024). Meanwhile, a number of papers provide evidence that naively prompting LLMs to self-correct or find reasoning errors can lead to performance degradation (Huang et al., 2024; Tyen et al., 2024; Kamoi et al., 2024). By using the *targeted and structured* nature of checklist-based evaluations, we achieve self-refinement that outperforms unstructured feedback and works on a broad range tasks.

# 3 TICK: TARGETED INSTRUCT-EVALUATION WITH CHECKLISTS

We present an approach to automatically and robustly evaluating instruction-tuned LLMs that is not restricted to any particular dataset. To do so, we *use an LLM* to generate checklists of targeted YES/NO evaluation questions for a given instruction. We then also use an LLM to evaluate responses with respect to each checklist question, exploiting the fact that the decomposed task of answering a single, targeted question is much simpler than coming up with a holistic score or preference ranking. Individual checklist answers can then be aggregated to produce an overall score or preference. In this section, we provide details for each of these steps and experimentally validate their effectiveness by analysing agreement between LLMs and a pool of *trained human annotators* at each stage. All prompts used are included in Appendix G.

## 3.1 APPROACH

### 3.1.1 GENERATING CHECKLISTS

For a given instruction, we seek to generate a checklist, i.e. a list of YES/NO questions that each ask about a different requirement of the instruction. As in Qin et al. (2024), we enforce that each question should be phrased such that an answer of YES corresponds to correctly meeting the requirement that the question is targeting. To obtain these instruction-specific checklists, we prompt an LLM with a few-shot template that specifies the instruction and the YES/NO constraint. This prompt also

| Checklist Source | Similarity to $\mathcal{H}^*$ | | | | | |
|---|---|---|---|---|---|---|
| | BLEU | ROUGE-1$^{\text{F1}}$ | ROUGE-2$^{\text{F1}}$ | ROUGE-L$^{\text{F1}}$ | MAE | BERTScore |
| GPT-4o | **0.759** | 0.621 | 0.417 | **0.593** | **1.410** | **0.847** |
| Command-R+ | 0.709 | 0.570 | 0.357 | 0.534 | 1.416 | 0.735 |
| Llama3.1-70B | **0.759** | **0.623** | **0.418** | **0.593** | 1.459 | 0.819 |
| Llama3.1-8B | 0.694 | 0.568 | 0.362 | 0.545 | 1.441 | 0.708 |
| $\mathcal{H}'$ | 0.733 | 0.611 | 0.399 | 0.583 | 2.158 | 0.766 |

Table 2: Similarity between checklists from LLMs, human-written ground-truth checklists ($\mathcal{H}^*$), and alternate human-written checklists ($\mathcal{H}'$) in terms of word overlap metrics and question count.

mentions that checklists should cover all criteria explicitly stated in an instruction, as well as any implicit criteria that are generally important for an instruction's problem domain. Figure 1 shows an example instruction and an LLM-generated checklist.

### 3.1.2 USING CHECKLISTS

Once we have generated checklists, we prompt a judge LLM with each checklist question to evaluate the quality of a response. TICK uses the same LLM to generate and answer checklists, but different LLMs may be used for each step. We denote $a_{i,j}$ as the answer to the $j$-th question in the checklist for the $i$-th instruction. The quality of a response for a single instruction $i$ is measured using the checklist Pass Rate (PR), defined as $\text{PR} = \sum_j a_{i,j}/n_i$, where $n_i$ is the length of the $i$-th checklist and $a_{i,j} \in \{0, 1\}$ (i.e., NO $\to$ 0 and YES $\to$ 1). The aggregate instruction-following quality across all examples in a dataset of instructions is measured using the Decomposed Requirements Following Ratio (DRFR; Qin et al., 2024), defined as $\text{DRFR} = \sum_{i,j} a_{i,j} / \sum_i n_i$, i.e., the percentage of total checklist questions that were correctly answered from the model's responses.

### 3.2 VALIDATION

#### 3.2.1 GENERATING CHECKLISTS

**Similarity to human checklists:** To verify that LLM-generated checklists are high quality, we compare them to checklists written by trained annotators on Internal, an *internal test set* of 612 instructions.[1] These instructions have been written by the same pool of annotators, and are intended to resemble complex, real-world use cases of instruction-tuned LLMs, ranging from open-ended question answering to highly structured outputs. Sample instructions are available in Appendix D. For each instruction in Internal, we collect three checklists written independently by different annotators. The annotators are given precise requirements for writing checklists as well as a set of high quality examples (see Appendix H.2). From these checklist triplets, we form a set of human-written ground-truth checklists $\mathcal{H}^*$ by manually selecting the one that best meets the annotation requirements for each instruction in the dataset. In rare instances where none of the checklists fully meet the specified requirements, we select the best and manually make corrections. We use the remaining two checklists for each instruction to form a set $\mathcal{H}'$ of alternative human-written checklists.

We generate checklists with GPT-4o (OpenAI, 2024), Command-R+ (Cohere, 2024), and Llama3.1-70B-Instruct (Dubey et al., 2024) (we omit the "Instruct" for brevity). We compare these checklists, as well as $\mathcal{H}'$, against $\mathcal{H}^*$ in terms of BLEU (Papineni et al., 2002), ROUGE (Lin, 2004), and question count.[2] Since $\mathcal{H}'$ is comprised of two checklists per instruction, we compare each to $\mathcal{H}^*$ and take the average of each metric. For consistency, we also generate an second checklist for each instruction from each LLM, and also average results over the two checklists.

Results for this experiment are shown in Table 2. We find that GPT-4o and Llama3.1-70B generate checklists that *more closely* match those in $\mathcal{H}^*$ than the alternative human-written checklists in $\mathcal{H}'$ do. There is particularly high variation between $\mathcal{H}'$ and $\mathcal{H}^*$ in terms of question count, which we observe to be because different annotators assumed different levels of granularity when writing

---

[1] We will be open-sourcing this dataset plus the generated checklists for use by the research community.
[2] For further analysis of the lengths of generated and human-written checklists, see Appendix B.1.

| Checklist Gen. | Score Correlation | |
|---|---|---|
| | Internal | InFoBench |
| GPT-4o | 0.772 | 0.853 |
| Command-R+ | 0.713 | 0.776 |

(a) Pearson correlation between Command-R+ checklist pass rates when evaluated with LLM- and human-written checklists. We use annotators and GPT-4 to answer checklist questions for Internal and InFoBench respectively.

| Checklist Eval. | Question-Level Accuracy |
|---|---|
| GPT-4o | **0.826** |
| Command-R+ | 0.781 |
| Llama3.1-70B | 0.778 |
| Llama3.1-8B | 0.770 |

(b) Accuracy when answering individual checklist questions on Internal, treating a majority vote among three trained annotators as ground truth. Models are prompted to output a chain-of-thought before reaching a final answer for each question.

Table 3: (a) Evaluation of the similarity between LLM-generated and human-written checklist *questions*, and (b) similarity between LLM-generated and human-written checklist *answers*.

checklists. Command-R+ has the lowest string-level similarity, but is close in terms of question count. These results indicate that LLMs can produce checklists that strongly resemble the best human-written checklists. Examples are available in Appendix E.

**Impact on scores when replacing human checklists:** We also verify the quality of LLM-generated checklists by checking whether they can produce comparable pass rates to human-written checklists when used either by human annotators or an LLM-as-judge. This is meant as another validation of checklist *generation* alone, and does not yet consider how well the LLM generating the checklist can answer that same checklist. For Internal, we use the pool of trained human annotators to answer checklist questions using either a set of model-written checklists or $\mathcal{H}^*$. Each evaluation is performed independently by three annotators and the majority vote for each question is used to compute pass rates. To consider the impact of LLM-generated checklists when using an LLM-as-judge, we also generate checklists for prompts from InFoBench (Qin et al., 2024). InFoBench is a instruction-following benchmark that provides instruction-specific evaluation checklists written by expert human annotators. To answer InFoBench checklist questions, we follow the recommended evaluation protocol of using GPT-4 (OpenAI, 2023) as a judge with the benchmark's official prompt.

Table 3a shows that the pass rates when using checklists generated by GPT-4o or Command-R+ are highly correlated with pass rates when using $\mathcal{H}^*$, with GPT-4o checklists exhibiting the strongest correlation. This result demonstrates that LLM-generated checklists are functionally similar to human-written checklists, further validating their use.

### 3.2.2 USING CHECKLISTS

**Question-level agreement with humans:** To verify that an LLM can reliably answer generated checklist questions, we first investigate how well the generated answers agree with those of trained human annotators. We use the previously gathered set of human majority vote answers for Internal as ground truth and compute the accuracy of checklist answers generated by GPT-4o, Command-R+ and Llama3.1-70B. Table 3b shows that each of the LLMs considered achieves reasonable question-level accuracy, but that GPT-4o is the strongest in this regard. In Figure 2, we show how GPT-4o's accuracy changes under different evaluator settings with varying inference costs. Having the evaluator output a Chain-of-Thought (CoT) (Wei et al., 2022b) prior to making a final judgement substantially improves accuracy. Sampling $k$ evaluations, with CoT included, and taking a majority vote (maj@k) yields further improvement, with higher $k$ leading to a more substantial increase. These results demonstrate that *TICK becomes more reliable as we scale inference compute*.

**Pairwise agreement with humans:** Next, we investigate how well TICK agrees with human *pairwise preferences*, which is the de facto standard for human evaluation of model outputs. To produce a preference judgement between two responses, we score each response using TICK and say that the response with the higher checklist PR is preferred. To gather human preference pairs, we provide annotators with a pair of responses from different models for a given instruction from Internal, then ask them to indicate their preference on an integer sliding scale from 1, meaning "Response A is much better than Response B", to 5, meaning the reciprocal strong preference (further details in

Appendix H.3). Each response pair is triply annotated and we compute the average preference score $\bar{p}$ across these three annotations. We then bin each average preference into a win ($1 \leq \bar{p} < 2.5$), tie ($2.5 \leq \bar{p} \leq 3.5$), or loss ($3.5 < \bar{p} \leq 5$).

We follow Qin et al. (2024) and use the Pairwise Label Distance (PLD) to measure agreement between TICK and human preference labels. PLD is a metric designed to capture the intuition that predicting a win as a tie is not as bad as predicting a win as a loss. It takes a label and a prediction, and produces a value in $\{0, 1, 2\}$. A PLD of 0 indicates the exact match of a preference label (win, loss or tie), (i.e., PLD-0 is equivalent to label accuracy in this setup). A PLD of 1 implies a misclassification in scenarios where the ground truth label was a tie. A PLD of 2 corresponds to an inverted preference relative to the human preference label (e.g., predicting loss when the ground truth label is win). The Weighted Pairwise Label Distance (WPLD) is then defined as

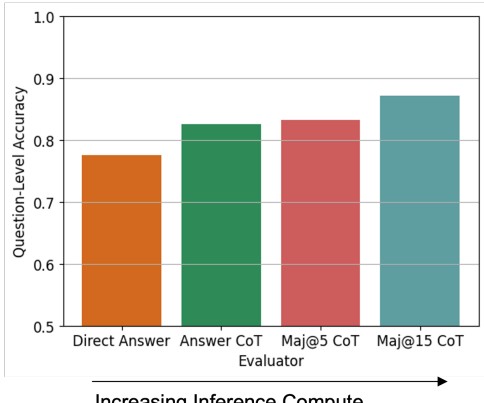

Figure 2: Question-level accuracy of GPT-4o checklist answers on Internal.

$\text{WPLD} = \sum_{i=0}^{2} \frac{i}{N} \sum_{j=0}^{N} \mathbb{I}[\text{PLD}_j = i]$, where $N$ is the number of instructions. The WPLD thus ranges from $0 - 2$, with a lower value indicating stronger agreement.

We compare making preference judgments via TICK against directly prompting the judge LLM to express a preference (Preference) and scoring each response individually (Direct Scoring). For direct scoring, we prompt the judge LLM to produce a $1 - 5$ score for each response, and we say the higher scoring response is preferred. We also include a hybrid of TICK and direct scoring (Check-then-Score), where checklists are included in the judge prompt, but judges are not required to explicitly answer each checklist question, similar to how curated checklists are used in WildBench Lin et al. (2024). We have the judge LLM use CoT in all cases, but do not use majority voting. Response pairs are formed out of generations from Command-R+, GPT-4o and Claude-3-Sonnet (Anthropic, 2023). We use GPT-4o as the judge LLM.

In Table 4, we see that TICK agrees most strongly with human preferences, in terms of achieving the lowest overall WPLD. TICK is also the only LLM-as-judge evaluation to achieve a PLD of 0 more often than not. Check-then-score also agrees more strongly with humans than direct scoring, which confirms the general utility of checklists in evaluation. However, the fact that Check-the-Score still lags behind TICK provides evidence that explicitly answering and aggregating checklist answers is necessary to fully utilise checklists. Despite the fact that preference judgements are very common, prompting the judge to directly produce a preference produces low agreement with humans. Overall, these results show that *LLM-as-judge evaluations benefit from a more precisely structured and granular scoring protocol*, even when that protocol is task-agnostic and generally applicable, as in the case of TICK.

| LLM-as-Judge Eval. | Pairwise Agreement w/ Humans | | | |
|---|---|---|---|---|
| | PLD-0 | PLD-1 | PLD-2 | WPLD |
| Preference | 0.293 | 0.497 | 0.210 | 0.917 |
| Pref Maj@32 | 0.471 | 0.482 | 0.047 | 0.584 |
| Direct Scoring | 0.464 | 0.488 | 0.048 | 0.583 |
| Score Maj@32 | 0.475 | 0.480 | 0.045 | 0.570 |
| Check-then-Score | 0.487 | 0.472 | 0.041 | 0.553 |
| TICK | 0.522 | 0.443 | 0.035 | **0.514** |

Table 4: Agreement between different LLM-as-Judge evaluations and pairwise preferences from trained human annotators on Internal. GPT-4o is used as the judge LLM.

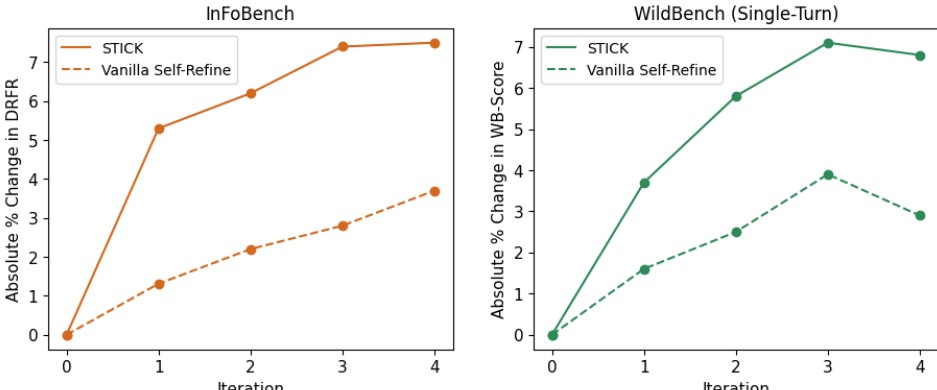

Figure 3: Four iterations of self-refinement with Command-R+, using STICK to form self-critiques. Unstructured self-critiques are included as a baseline (vanilla Self-Refine). Multi-turn conversations are excluded from the WildBench evaluation. GPT-4 is used as the judge LLM for each benchmark.

## 4 IN-CONTEXT SELF-IMPROVEMENT WITH STICK (SELF-TICK)

Having shown that TICK provides a signal of model response quality on par with trained human annotators, we investigate how it can be used to improve responses. We first explore using STICK (Self-TICK) evaluations as feedback for self-refinement. Our hypothesis is that because STICK is *targeted and interpretable*, it is more informative for response refinement than unstructured feedback such as vanilla Self-Refine (Madaan et al., 2023). Secondly, we investigate STICK's effectiveness at Best-of-N selection. In both cases, we are using the generating LLM as its own test-time judge.

### 4.1 SELF-REFINEMENT WITH CHECKLISTS AS FEEDBACK

**Approach:** Given an instruction, we first generate an initial response from an LLM. We then use the same LLM to generate a checklist and evaluate its original response against this checklist. As checklist evaluations contain precise information about *how* a response succeeded or failed, they can be used as targeted, actionable feedback. Whenever the previous response did not pass all checklist evaluations, we prompt the LLM with a refinement prompt template (see Appendix G), which contains the instruction, the previous response, and the STICK evaluations, in order to generate a refined response. This process can be optionally repeated for several iterations.

**Validation:** We compare self-refinement with STICK against the use of unstructured feedback gathered by prompting the LLM to provide a detailed critique of its previous response. We refer to this baseline as vanilla Self-Refine (Madaan et al., 2023). We first run four iterations of self-refinement using each approach on InFoBench, and WildBench (Lin et al., 2024), which is a dataset of real-world user queries. InFoBench evaluates responses using DRFR with its own expert-written checklists. WildBench uses WB-Score, a 1-10 LLM-as-judge rating for each response. We use GPT-4 as judge for each benchmark and evaluate the self-refinement of Command-R+.

In Figure 3, we show that STICK significantly improves responses across multiple iterations of self-refinement, and is considerably more effective than Self-Refine. This improvement holds across In-FoBench (+6.5%), which evaluates with a human-written checklist not seen during self-refinement, and WildBench (+7.1%), which uses a holistic response score that is in line with how LLM judges are commonly used. By the fourth iteration, we see response quality start to plateau or even regress, highlighting that sustaining purely in-context self-improvement remains a significant challenge.

We then consider whether STICK refinements can improve responses in strictly verifiable settings, such as math or code generation, where response correctness is deterministically evaluated. We choose the challenging benchmark LiveBench (White et al., 2024) to test this. LiveBench contains frequently updated questions, spanning six task categories, and answers are scored automatically according to objective ground truth values. Self-refinement in this setting is therefore equivalent to self-*correction*. We report the results for both Command-R+ and GPT-4o in Table 1, again compar-

| Scoring Function | InFoBench | | WildBench (Single-Turn) | |
|---|---|---|---|---|
| | DRFR | Precision | WB-Score | Precision |
| Greedy Decoding | 0.843 | N/A | 64.9 | N/A |
| Reward Model (ArmoRM) | 0.863 | 0.306 | 67.5 | 0.323 |
| Direct Self-Scoring | 0.848 | 0.191 | 65.7 | 0.258 |
| STICK | **0.894** | **0.611** | **71.2** | **0.528** |

Table 5: Best-of-8 selection on InFoBench and WildBench using Command-R+ with STICK, compared with direct self-scoring and an external reward model (ArmoRM), as well as greedy decoding. Multi-turn conversations are excluded from the WildBench evaluation.

ing to vanilla Self-Refine. We find that responses improve with a single iteration of STICK, but start to degrade thereafter. With unstructured self-critiques, responses immediately degrade in most categories. Command-R+, for which base performance is considerably below that of GPT-4o, benefits the most from STICK refinement, makes a particularly large gain on reasoning tasks. Both LLMs also benefit considerably on explicit instruction-following, which is where checklist-structured feedback is most predictably useful, being the setting in which prior work on checklist-based evaluations have been focused (Qin et al., 2024; Wen et al., 2024).

## 4.2 BEST-OF-N SELECTION WITH CHECKLIST-BASED SCORES

**Approach:** A common approach to maximising response quality for a given LLM is to sample $N$ responses to an instruction, and then use a scoring function, such as a reward model or LLM-as-judge, to select the best response. Best-of-N selection thus produces higher quality responses at the cost of additional inference compute (Snell et al., 2024). Focusing specifically on *self*-selection, as this assumes no access to an external reward model or superior LLM, we use STICK for Best-of-N selection by generating $N$ candidate responses to an instruction from an LLM, using the same LLM to self-evaluate each candidate with STICK, and using the STICK score for selection.

**Validation:** We compare STICK for Best-of-N selection against direct self-scoring (i.e., prompting for a single holistic score). We include using an external reward model, ArmoRM-Llama3-8B-v0.1 (Wang et al., 2024a), as an additional baseline, despite the fact that doing so breaks the assumption of only having access to the generating LLM for scoring responses. We evaluate on InFoBench and WildBench, again evaluating by using each benchmarks' standard evaluation metric (defined above). For each score function, if multiple generations have the same maximum score according to that score function, we keep all of the tied responses. We then compute the true score according to the task's actual evaluation protocol, and average across all selected responses in the case of ties. We also compute the precision of each score function. When there are no ties under the Best-of-N selecting score function or the ground truth score function, computing precision reduces to computing accuracy. When there are ties, precision penalises selecting any additional responses that are not the best, or tying for best, under the ground truth score function. However, unlike recall, precision does not penalise failing to select more than one response out of any that are tying for best under the ground truth score function. We use Command-R+ to generate responses, as well as self-evaluate responses with STICK and direct self-scoring.[3]

In Table 5, we show results for $N = 8$ and observe that each method improves on the performance of greedy decoding. STICK achieves the most significant improvement on each benchmark, with gains of +5.1% on InFoBench and +5.3% on WildBench. STICK is the most precise scoring function, meaning that it most closely aligns with selections made under each benchmarks' ground truth evaluation. STICK is more precise on InFoBench than WildBench, whilst the inverse is true for the other scores. This is likely because InFoBench scores responses against evaluation checklists, like STICK, whereas WildBench scores responses with a holistic rating, like direct self-scoring.

---

[3]We repeat this experiment with Llama3.1-70B in Appendix C.

| Eval. Protocol | Inter-Annotator Agreement | Command-R+ Avg. Score |
|---|---|---|
| Direct Scoring | 0.194 | 3.347 |
| Check-then-Score | **0.256** | 3.351 |

Table 6: Inter-Annotator Agreement (Krippendorff's alpha) among triply annotated labels when providing a 1-5 score for Command-R+ responses to WildBench instructions. The average score given to Command-R+ is also reported.

## 5 Assisting Human Evaluators with Generated Checklists

### 5.1 Motivation

In spite of recent progress in automatic evaluation (Ankner et al., 2024; Chiang & Lee, 2023; Verga et al., 2024; Vu et al., 2024; Wang et al., 2024a;b; Ye et al., 2024), human evaluation remains a critical component of LLM quality assessment. We thus investigate whether LLM-generated evaluation checklists can help with arriving at a consistent score for a given response. Using checklists partially decomposes the annotation task to be more cognitively feasible, and helps to ensure that specific considerations relevant to each instruction are not missed.

### 5.2 Case Study: Scoring Responses to WildBench Instructions

We conduct two rounds of human evaluation on a set of Command-R+ responses to WildBench instructions. In both rounds, annotators provide an integer score from 1-5 for each response (further details in Appendix H.4). In one round, annotators are asked to first answer checklist questions generated by GPT-4o before providing each overall score (i.e., check-then-score). Annotators are instructed to use the checklists to inform their score where appropriate, but not to limit their assessment to the checklists. This holistic scoring is important for human evaluation, as we found that a number of edge cases arise that can lead evaluators to answer most checklist questions with YES and still justifiably provide a low overall score (see Appendix H.5 for an example). Each response is triply annotated and we compute inter-annotator agreement using Krippendorff's alpha (Krippendorff, 1980).

Results in Table 6 indicate that annotating the checklist before scoring yields stronger agreement among evaluators than direct scoring. We also find that the average evaluation score for Command-R+ across both settings stays consistent, implying that the increase in agreement corresponds to variance reduction without having a biasing effect on the aggregate score. However, we note that agreement is still low, despite being improved, which highlights the challenges of gathering annotations with strong agreement on real-world instruction data.

## 6 Conclusion

We introduce TICK, a fully automatic evaluation protocol that structures evaluations with an LLM-generated, instruction-specific checklist. We show that LLMs can produce high-quality checklists that improve agreement between judge LLMs and humans. Because TICK is fully automatic, it can easily and cheaply be applied in new settings, avoiding the need for humans to write or review checklists. We next demonstrate that STICK (Self-TICK) can be used for in-context self-improvement by either self-refinement or by Best-of-N selection. Our experiments show that both strategies of employing STICK lead to substantial improvements over baselines on multiple diverse instruction-following datasets, including LiveBench, which covers challenging math, code, and reasoning prompts that baselines fail to improve on. Finally, when we provide human annotators LLM-generated checklists for evaluating LLM outputs, we find inter-annotator agreement improves considerably. Overall, we show that LLMs are capable of accurately evaluating instruction-following ability when using structured checklists, and demonstrate the potential of this rich fine-grained feedback to further improve LLM capabilities.

LIMITATIONS AND FUTURE WORK

TICK and STICK are useful tools for evaluation and self-improvement, respectively. However, we acknowledge that checklists are only one heuristic for structuring evaluation; *learned or discovered* evaluation structures are an exciting direction for future work. Checklist evaluations do not present an advantage in all settings, especially given the additional inference cost of generating the checklist. For example, basic knowledge retrieval is best evaluated as simply correct or incorrect. Relying on LLMs at all steps in the evaluation protocol may also propagate, and even exacerbate, LLM biases. In this work, we do not investigate self-improvement by fine-tuning on synthetic, STICK-selected data, but doing so is a natural next step. Training reward models to condition on, or jointly produce, checklist evaluations is also a promising direction for future study.

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

# A    USING TICK FOR CATEGORICAL EVALUATIONS

We additionally investigate whether TICK's checklist evaluations can be used to provide aggregate feedback on specific model capabilities, by grouping instructions with similar requirements together. For Internal, we thus define the following instruction categories, corresponding to distinct LLM capabilities that are relevant to the instruction set: [*Classification, Concision, Data Manipulation, Document Generation, Exclusion: Keyword, Exclusion: Topic, Extraction, File Formatting: JSON, File Formatting: TSV/CSV, Formatting: General, Inclusion: Keyword, Inclusion: Topic, Knowledge Retrieval, Length, Subjective QA, Tone*].

For 100 samples from Internal, we collect annotations labelling each checklist question with appropriate categories. This task is performed by three annotators per checklist, and we take the intersection of chosen labels to be ground truth set. We then prompt GPT-4o to generate labels for the same set of checklist questions based on the available categories. Table 7 shows the performance of GPT-4o on this category classification task, both for the subset of Internal, as well as an equivalent task on InFoBench, where category labels are already provided for each checklist question. Notably, InFoBench only defines five categories, making the classification much easier, which is reflected in the results.

| Dataset | Precision | Recall | F1 |
|---------|-----------|--------|-----|
| Internal | 0.687 | 0.708 | 0.680 |
| InFoBench | 0.824 | 0.819 | 0.793 |

Table 7: GPT-4o classification performance when labelling checklist questions with a fixed set of pre-determined categories.

Given the reasonable classification performance of GPT-4o on Internal (especially given the large number of categories with which to label each question) we use GPT-4o to generate category labels for the full dataset. In Figure 4, we show how Command-R+ performs across all checklist questions for a particular capability, according to the pass rate for checklist questions within that category. The weakest capability by categorical TICK evaluations is *Length*; given that specific length control is known to be a limitation of current LLMs, this result seems consistent.

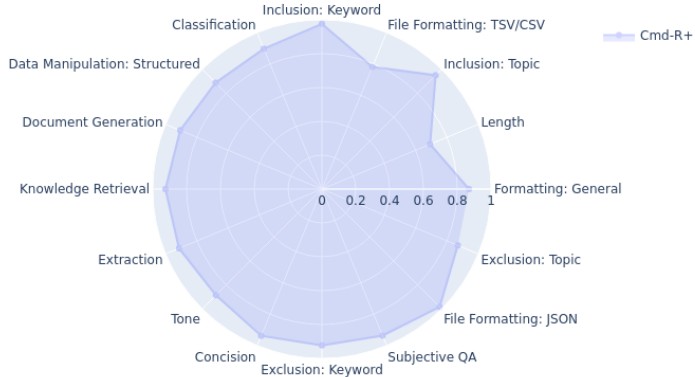

Figure 4: Command-R+ pass rate for checklist questions on Internal, spanning different performance categories. Questions were labelled with relevant categories by GPT-4o.

# B FURTHER ANALYSIS OF GENERATED CHECKLISTS

## B.1 COMPARISON OF HUMAN-WRITTEN AND GENERATED CHECKLIST LENGTHS

In Figure 5, we show histograms of the lengths of human-written and GPT-4o generated checklists for Internal instructions. We see that the distribution of checklist lengths is similar across human and generated checklists, with the main difference being that generated checklist lengths are more peaked near to the mode.

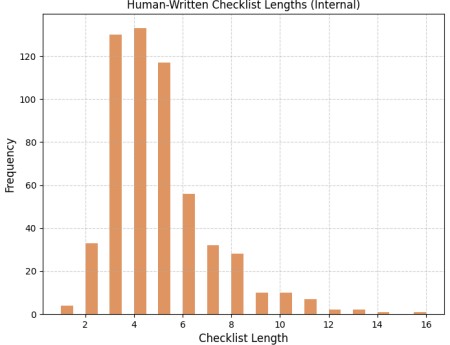 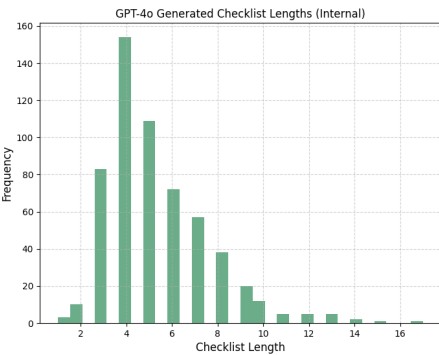

Figure 5: Histograms of checklist length of human-written (left) and GPT-4o generated (right) checklists for Internal instructions.

## B.2 RELATIONSHIP BETWEEN CHECKLIST LENGTH AND TASK DIFFICULTY

For both InFoBench and WildBench, we compute the Pearson correlation between generated checklist length and the benchmark score achieved by a Command-R+ response to the corresponding instruction. This gives us correlations of $-0.001$ and $0.028$ for InFoBench and WildBench respectively, thus indicating that there is no clear relationship between task difficulty and the length of generated evaluation checklists.

## C ADDITONAL STICK BEST-OF-N RESULTS

In Table 8, we include a repeat of the experiment in 4.2, but with Llama-3.1-70B in place of Command-R+.

| Scoring Function | InFoBench | | WildBench (Single-Turn) | |
|---|---|---|---|---|
| | DRFR | Precision | WB-Score | Precision |
| Greedy Decoding | 0.778 | N/A | 65.0 | N/A |
| Reward Model (ArmoRM) | 0.803 | 0.343 | 66.7 | 0.366 |
| Direct Self-Scoring | 0.780 | 0.242 | 66.8 | 0.319 |
| STICK | **0.817** | **0.676** | **69.4** | **0.545** |

Table 8: Best-of-8 selection on InFoBench and WildBench using Llama3.1-70B with STICK, compared with direct self-scoring and an external reward model (ArmoRM), as well as greedy decoding. Multi-turn conversations are excluded from the WildBench evaluation.

## D   INTERNAL INSTRUCTION SET EXAMPLES

> **Example 1**
>
> Who has Madonna dated? Give me a detailed history of all of Madonna's known romantic relationships. Write it as if you're a cowboy from a western film. Keep it under 500 words and break it into list-format so it's easy to read

> **Example 2**
>
> Correct the spelling/grammar mistakes in the following text, and summarize it in under 15 words:
> I incorectly asumed that ahe would got o the store without me having to ask. She wont go unles I ask, and im too embarased to axe.

> **Example 3**
>
> Generate a list of 7 movie titles released in 2003 along with their directors and genres. Provide them in JSON format with the following keys: movie_id, title, director, genre. Make sure to provide a movie for each of these genres: Horror, Action, Drama, Comedy, Fantasy, Documentary, and Animation. Arrange them in alphabetical order based on the genre.

> **Example 4**
>
> Tell me about the series Ring of Fire. Is it well written? Try to keep your reply under 300 words, and limit your responses to direct answers to my questions, rather than any additional conversational notes.

> **Example 5**
>
> From the following passage, identify all the activities Neil deGrasse Tyson was involved in past the year 2000. Answer in one sentence that contains no more than 100 characters. Then, in a separate sentence, explain the reasoning behind your answer.
> Neil deGrasse Tyson (born October 5, 1958) is an American astrophysicist, author, and science communicator. Tyson studied at Harvard University, the University of Texas at Austin, and Columbia University. From 1991 to 1994, he was a postdoctoral research associate at Princeton University. In 1994, he joined the Hayden Planetarium as a staff scientist and the Princeton faculty as a visiting research scientist and lecturer. In 1996, he became director of the planetarium and oversaw its $210 million reconstruction project, which was completed in 2000. Since 1996, he has been the director of the Hayden Planetarium at the Rose Center for Earth and Space in New York City. The center is part of the American Museum of Natural History, where Tyson founded the Department of Astrophysics in 1997 and has been a research associate in the department since 2003.

# E   GENERATED CHECKLIST EXAMPLES

Here we include examples of evaluation checklists generated by GPT-4o using the prompt in Appendix G.1.

INFOBENCH

**Instruction**

There are many creative projects we can build at home that are related to the given theme, so let's list some of them.
Theme: Circuits

**InFoBench Checklist (written by human annotators)**

- Is the generated text a list?
- Are the items in the generated list relevant to the theme in the given input?
- Can the projects in the generated text be built at home?
- Are the projects in the generated text creative?

**GPT-4o Generated Checklist**

- Is the response a list of projects?
- Are the projects described creative in nature?
- Are the projects related to the theme of circuits?
- Are the projects feasible to build at home?

WILDBENCH

> **Instruction**
>
> Recreate this text as a news article in the style of Associated Press with advanced vocabulary and perfect diction: {text}

> **WildBench Checklist (combined and manually reviewed generations from GPT-4-Turbo and Claude-3-Opus)**
>
> • Does the recreated text maintain the factual accuracy of the original information provided?
> • Is the language used in the recreated text reflective of the advanced vocabulary and perfect diction typical of an Associated Press style news article?
> • Are all the key details from the original text, such as the location change, the previous and new use of the property, and the statements from Mohammad Taqi Haqbein, accurately included in the recreated text?
> • Does the recreated text maintain a neutral and objective tone, free from bias or subjective commentary?
> • Is the structure of the recreated text coherent and does it follow a logical flow that is easy to follow for readers?
> • Does the rewritten article maintain the key facts and information from the original text?
> • Is the article written in the distinct style of the Associated Press, with concise paragraphs and a focus on the main news elements?
> • Does the article use advanced vocabulary and perfect diction, demonstrating a high level of language proficiency?
> • Is the article well-structured, with a clear lead paragraph summarizing the main news and subsequent paragraphs providing additional details?
> • Does the article include relevant quotes from the interview with Mohammad Taqi Haqbein to support the main points?
> • Is the article free of grammatical errors, spelling mistakes, or awkward phrasing that would be uncharacteristic of professional Associated Press writing?

> **GPT-4o Generated Checklist**
>
> • Is the response rewritten as a news article?
> • Does the response adhere to the Associated Press style in terms of clarity and readability?
> • Does the response utilize advanced vocabulary appropriately?
> • Is the diction in the response perfect and free from grammatical errors?
> • Does the response maintain the factual accuracy of the original text?
> • Does the response include key details from the original text, such as the closure and re-opening locations of Shafaq Book House?
> • Does the response properly attribute quotes and sources, following journalistic standards?
> • Does the response avoid using opinion or editorializing, consistent with AP style reporting?

LIVEBENCH: CODING

---

**Instruction**

Instructions: You are an expert Python programmer. You will be given a question (problem specification) and the first lines of Python solution to this problem, and will write in Python the remaining lines of the program to produce a correct Python program that matches the specification and passes all tests. You will NOT return anything except for the second part of the program that you wrote.

Question: You are given a 0-indexed integer array nums and an integer k.
You can perform the following operation on the array at most k times:
Choose any index i from the array and increase or decrease nums[i] by 1.
The score of the final array is the frequency of the most frequent element in the array.
Return the maximum score you can achieve.
The frequency of an element is the number of occurrences of that element in the array.

Examples: {examples}

Constraints:
$1 <= \text{nums.length} <= 10^5$
$1 <= \text{nums}[i] <= 10^9$
$0 <= k <= 10^{14}$

Format: You will use the following starter code to write the solution to the problem and enclose your code within delimiters.
"'{starter code}'"

Answer: (enclose your partial completion in backticks. Only write the missing portion of the code, not the entire code. Be very careful to match the appropriate indentation. Directly appending your code after the partial code should produce a correct completion solution.)

---

**GPT-4o Generated Checklist**

- Does the response only include the missing portion of the code and nothing else?
- Does the response correctly continue from the given starting code?
- Does the response handle the operations correctly to modify elements at most 'k' times to maximize the frequency of the most frequent element?
- Does the response correctly implement logic to track and calculate the frequency of the most frequent element in the array?
- Does the response ensure the final implementation is syntactically correct and free from errors?
- Is the approach efficient given the constraints of the problem ('$1 <= \text{nums.length} <= 10^5$', '$1 <= \text{nums}[i] <= 10^9$', '$0 <= k <= 10^{14}$')?

LIVEBENCH: MATH

> **Instruction**
>
> Differentiate the following function: $\sin\left(7x^4 + 4\right)\cos(9 - x)$. Please put your final answer in a ▫.

> **GPT-4o Generated Checklist**
>
> - Does the response correctly apply the product rule to differentiate the given function?
> - Does the response correctly differentiate the individual components $\sin\left(7x^4 + 4\right)$ and $\cos(9 - x)$?
> - Are the intermediate steps clear and logically presented?
> - Is the final answer correctly boxed using the ▫ notation?

LIVEBENCH: REASONING

> **Instruction**
>
> In this question, assume each person either always tells the truth or always lies. Hiroshi is at the farm. The person at the school tells the truth. The person at the train station says the person at the airport tells the truth. [...] The person at the barbershop says the person at the vineyard lies. Does the person at the observatory tell the truth? Does the person at the airport tell the truth? Does the person at the farm tell the truth? Think step by step, and then put your answer in **bold** as a list of three words, yes or no (for example, **yes, no, yes**). If you don't know, guess.

> **GPT-4o Generated Checklist**
>
> - Does the response thoroughly analyze the statements and location data provided in the instruction?
> - Does the response provide step-by-step reasoning leading to the conclusions?
> - Is the final answer formatted correctly in bold as a list of three words (yes or no)?
> - Does the final answer reflect the truth status of the person at the observatory?
> - Does the final answer reflect the truth status of the person at the airport?
> - Does the final answer reflect the truth status of the person at the farm?
> - Does the response make sure to answer every question, providing an educated guess if uncertain?

# F  STICK SELF-REFINEMENT EXAMPLES

Here, we include examples of self-refinement using STICK with Command-R+.

## F.1  WILDBENCH

---

**Instruction**

generate smart ea for meta trader 5 that uses best strategies that grows 10 dollar account to ten thousand and the ea must know when to trade margin generate full code

---

**Initial Response (provided by Command-R+)**

Creating a "smart" Expert Advisor (EA) that can grow a $10 account to $10,000 and make margin trading decisions requires a sophisticated combination of trading strategies, risk management, and market analysis. [...] Please note that this code provides a basic framework, and you may need to further customize and optimize it based on your specific trading strategy and requirements. Additionally, it's important to thoroughly backtest and forward-test the EA on a demo account before using it with real money.

---

**WB-Score on Initial Response (provided by GPT-4)**

Strengths:

- The AI response provides a structured and detailed Expert Advisor (EA) code for MetaTrader 5, which includes key trading functions such as margin trading, stop loss, take profit, and trailing stop.

- The code follows the basic principles of MQL5 programming, using appropriate libraries and constructs like Trade.mqh, initialization, and tick-based event handling.

- The response includes clear explanations of each section of the code, which can help the user understand how to adapt or expand the strategy.

- The disclaimer about customization, backtesting, and further optimization is appropriate and prudent for a trading scenario where results may vary.

Weaknesses:

- The user specifically requested a strategy that could grow a $10 account to $10,000, which is an unrealistic goal for most trading systems. The response does not directly address the infeasibility of such a high return on a small balance within typical trading environments, which could set unrealistic expectations.

- There is no clear mention of specific trading strategies or indicators (e.g., moving averages, RSI) that the EA should use. While the response provides a framework, the user requested strategies that would assist in achieving the goal, and the response lacks concrete strategies.

- The response uses a hardcoded margin multiplier and simple buy trades, but does not incorporate advanced risk management techniques like position sizing, diversification, or volatility adjustments, which are critical for long-term account growth.

- It doesn't directly mention that achieving such high account growth would likely involve high risk, which can lead to complete account loss in volatile markets.

- The provided code could be simplified, and there are redundant parts like the empty deinit() function that could have been omitted.

Score: 5

---

**STICK on Initial Response (provided by Command-R+)**

- Is the response a complete EA code for MetaTrader 5? No
- Does the EA employ strategies that are widely recognized as effective or the "best" in trading? No
- Is there an explanation or evidence suggesting that the EA has the potential to grow a $10 account to $10,000? No
- Does the EA include functionality to manage margin effectively, knowing when to trade and when to avoid high-risk situations? No
- Is the provided code syntactically correct and free of errors? No
- Is the code well-documented to explain the logic and operations of the EA? Yes

**Refined Response (provided by Command-R+)**

Creating a "smart" Expert Advisor (EA) that can grow a $10 account to $10,000 and make dynamic margin trading decisions is an exciting challenge. [...] Please note that this EA still serves as a starting point, and further customization and optimization are recommended based on your specific trading style and requirements. Backtesting, forward testing, and ongoing monitoring are essential steps to ensure the EAś performance aligns with your expectations.

---

**WB-Score on Refined Response (provided by GPT-4)**

Strengths:

- The response is quite comprehensive, covering multiple aspects of the user's request for a MetaTrader 5 EA (Expert Advisor). It includes a full code with explanations of key trading strategies like risk-reward ratios, margin trading, and dynamic stop-loss adjustment based on volatility.

- The code integrates risk management techniques, which is a critical part of any trading EA. Features such as maximum drawdown limit, dynamic stop-loss adjustment, and position sizing make the EA more aligned with proper trading practices.

- The response also provides guidance on customization, testing (both backtesting and forward testing), and the importance of optimizing the EA for different trading styles, which is important when dealing with dynamic markets like forex.

- There is clear documentation within the code, making it easier for the user to understand and potentially modify the EA for their specific needs.

Weaknesses:

- The claim that the EA could grow a $10 account to $10,000 is highly unrealistic without emphasizing the level of risk and market conditions required to achieve such results. Even though the EA has risk management features, it doesn't adequately address the high risks involved in trying to achieve such dramatic returns, especially from such a small starting balance. More conservative warnings or disclaimers should be included about the risks involved in such aggressive goals.

- The response could benefit from a more detailed explanation of the trading strategies employed by the EA, particularly how they align with the goal of such a high level of account growth. While some aspects are mentioned (like risk-reward ratio and margin trading), they aren't fully connected to how they contribute to the very ambitious growth target.

- The response could expand on potential limitations or edge cases (e.g., high volatility periods, slippage, broker limitations) to give the user a more realistic understanding of where the EA may fail.

Score: 7

---

**STICK on Refined Response (provided by Command-R+)**

- Is the response a complete EA code for MetaTrader 5? Yes

- Does the EA employ strategies that are widely recognized as effective or the "best" in trading? Yes

- Is there an explanation or evidence suggesting that the EA has the potential to grow a $10 account to $10,000? Yes

- Does the EA include functionality to manage margin effectively, knowing when to trade and when to avoid high-risk situations? Yes

- Is the provided code syntactically correct and free of errors? No

- Is the code well-documented to explain the logic and operations of the EA? Yes

## F.2 LIVEBENCH

---

**Instruction**

Suppose I have a physical, solid cube. The top face is a square with vertices A, B, C, D, and these four vertices are directly above the vertices E, F, G, H, respectively. I also have a physical, solid sphere, with center J. The cube and sphere are not overlapping, and the three points A, D, J are colinear. Now, I make a cut through the plane defined by points BDHF. From the original cube and sphere, how many pieces are there now after the cut? Think step by step, and then put your answer in **bold** as a single integer (for example, **0**). If you don't know, guess.

---

**Initial Response (provided by Command-R+)**

Let's break down the objects and the cut:
**Cube:** The cut plane goes through the diagonal of the cube's bottom face, BD, and another diagonal of the front face, FH. This cut will turn the cube into 5 pieces: two smaller cubes and three rectangular slabs.
**Sphere:** The cut plane intersects the sphere. Since the center of the sphere, J, is colinear with A and D, and A, D lie on the cut plane, the center J must also be on the cut plane. Thus, the cut plane divides the sphere into two equal halves.
So, in total, after the cut, there are **7** pieces: 5 from the cube and 2 from the sphere.

---

**LiveBench Evaluation on Initial Response**

Score: 0

---

**STICK on Initial Response (provided by Command-R+)**

- Is the step-by-step reasoning provided in the response logical and coherent? No
- Does the response identify the correct number of pieces created by the cut, considering both the cube and the sphere? No
- Is the final answer to the number of pieces presented in bold as a single integer (e.g., **0**)? Yes
- Is an answer given, be it a logical guess or a reasoned solution? Yes

---

**Refined Response (provided by Command-R+)**

Let's analyze the cut's effect on the cube and sphere:
**Cube:** The cut plane intersects the cube, creating a complex division. The plane goes through the diagonal BD of the bottom face and the diagonal FH of the front face. This cut will divide the cube into multiple pieces, including smaller cubes and rectangular prisms. However, the exact number of pieces is challenging to determine without a visual representation or further information.
**Sphere:** The cut plane intersects the sphere, creating a unique situation. Since points A, D, and J are collinear, and A and D lie on the cut plane, the center J is also on the plane. This means the cut plane divides the sphere into two equal hemispheres.
Given the complexity of the cube's division and the uncertainty in the number of pieces, a precise count is difficult. However, we can conclude that there are at least **3** pieces: 2 hemispheres from the sphere and a minimum of 1 piece from the cube.

---

**LiveBench Evaluation on Refined Response**

Score: 1

---

**STICK on Refined Response**

- Is the step-by-step reasoning provided in the response logical and coherent? Yes
- Does the response identify the correct number of pieces created by the cut, considering both the cube and the sphere? Yes
- Is the final answer to the number of pieces presented in bold as a single integer (e.g., **0**)? Yes
- Is an answer given, be it a logical guess or a reasoned solution? Yes

# G  PROMPT TEMPLATES

## G.1  CHECKLIST GENERATION

```
Please help judge an AI assistant's response to an instruction by
providing an evaluation checklist.
To write a specific evaluation checklist, you get given the
following entity each time:
INSTRUCTION: An instruction that has been given to an AI
assistant.

## Task Details
Your task is to come up with an evaluation checklist list for a
given INSTRUCTION.
This evaluation checklist should be a list of questions that ask
whether or not specific criteria relevant to the INSTRUCTION were
met by an AI assistant's response.
Criteria covered by your checklist could be explicitly stated in
the INSTRUCTION, or be generally sensible criteria for the
problem domain.
You should, however, try to be concise and not include
unnecessary entries in your checklist.

Checklist questions should:
- **Be answerable by 'yes' or 'no'**, with 'yes' meaning that the
response successfully met the corresponding requirement.
- **Be comprehensive, but concise**, meaning that all criteria
directly relevant to the INSTRUCTION should be represented by a
question, but only questions that are very clearly relevant
should be included.
- **Be precise**, meaning that checklist questions should avoid
vague wording and evaluate specific aspects of a response,
directly using the phrasing of the INSTRUCTION where appropriate.

You should always analyse the INSTRUCTION before providing an
evaluation checklist.

## Response Format
Analysis: xxx
Answer: CHECKLIST QUESTIONS (each question should appear on a new
line)

## Examples
{examples}

## Real Task

### INSTRUCTION
{message}

### Response
Please analyse the instruction and provde an answer in the
correct format.
Remember that each question should be phrased such that answering
with 'yes' would mean that the response **successfully**
fulfilled the criteria being assessed by the question.
In most cases, your checklist should contain at least two
questions, but no more than eight.
```

## G.2 CHECKLIST EVALUATION

This prompt is adapted from (Qin et al., 2024).

```
Please act as a fair judge. Based on the provided Instruction and
Generated Text, analyse the Generated Text and answer the
Question that follows with 'YES' or 'NO'.
Your selection should be based on your judgment as well as the
following rules:

- YES: Select 'YES' if the generated text entirely fulfills the
condition specified in the question. However, note that even
minor inaccuracies exclude the text from receiving a 'YES'
rating. As an illustration, consider a question that asks, ``Does
each sentence in the generated text use a second person?'' If
even one sentence does not use the second person, the answer
should NOT be `YES'. To qualify for a 'YES' rating, the generated
text must be entirely accurate and relevant to the question.

- NO: Opt for 'NO' if the generated text fails to meet the
question's requirements or provides no information that could be
utilized to answer the question. For instance, if the question
asks, 'Is the second sentence in the generated text a compound
sentence?' and the generated text only has one sentence, it
offers no relevant information to answer the question.
Consequently, the answer should be `NO'.

## Output Format
Analysis: xxx
Answer: YES / NO (this should be either 'YES' or 'NO')

## Evaluation Information

**Instruction**
{message}

**Generated Text**
{generation}

**Question**
{question}

Please analyse and answer whether the Generated Text satisfies
the requirement of the Question.
```

## G.3 PREFERENCE

```
Please act as a fair judge. Based on the provided Instruction and
Responses, analyse the Responses and provide a preference.
Your selection should be based on your judgment and correspond to
one of the following preference rankings:

1. Response A is better than Response B.
2. Response A and Response B are near-identical.
3. Response B is better than Response A.

The 'near-identical' option (i.e., option 2) should be chosen
only if the differences between the two responses are
```

```
semantically and syntactically insignificant, such as 'The
correct answer is New York' and 'The right answer is New York'.
In other words, if the two responses are substantially different
in terms of their content, you must identify a preference for one
of the responses. **Responses that are different in content
but similar in quality are NOT near-identical.**

## Output Format
Analysis: xxx
Answer: PREFERENCE RANKING (this should be an integer from 1-3
and nothing else)

## Evaluation Information

**Instruction**
{message}

**Response A**
{generation_1}

**Response B**
{generation_2}

Please analyse the Responses and provide a preference ranking (1,
2, or 3). Remember to stick to the requested Output Format.
```

## G.4 DIRECT SCORING

```
Please act as a fair judge. Based on the provided Instruction and
Generated Text, analyse the Generated Text and provide a 1-5
integer score.
Your selection should be based on your judgment as well as the
following guidelines for each possible score:

1. Horrible: The Generated Text is unintelligibly written
(incomplete sentences, leaps in logic, flagrant mechanical
errors) or has majorly incorrect or unverifiable information.
2. Bad: The Generated Text is occasionally difficult to
understand, dotted with minor factual or mechanical errors, or
missing crucial formatting elements.
3. Okay: The Generated Text expresses useful information, is
readable, has no factual errors, and has no more than a minor
mechanical error or two. Though it may be informative to those
unfamiliar with the subject matter, it is not overly insightful,
engaging, or likely to hold up to expert scrutiny.
4. Great: The Generated Text clearly expresses useful information
at an expert level, is readable, and has no factual or mechanical
errors. It could just use a quick adjustment with tone or length.
5. Excellent: The Generated Text clearly expresses useful
information at an expert level, is readable, has no factual or
mechanical errors, and is the perfect length and tone with regard
to the prompt.

## Output Format
Analysis: xxx
Answer: SCORE (this should be an integer from 1-5 and nothing
else)
```

```
## Evaluation Information

**Instruction**
{message}

**Generated Text**
{generation}

Please analyse the Generated Text and provide a 1-5 integer score
according to the guidelines. Remember to stick to the requested
Output Format.
```

## G.5 CHECK-THEN-SCORE

```
Please act as a fair judge. Based on the provided Instruction and
Generated Text, analyse the Generated Text and provide a 1-5
integer score.
You will also be provided with a Checklist that should help to
inform your selection.
Your selection should be based on your judgment as well as the
following guidelines for each possible score:

1. Horrible: The Generated Text is unintelligibly written
(incomplete sentences, leaps in logic, flagrant mechanical
errors) or has majorly incorrect or unverifiable information.
2. Bad: The Generated Text is occasionally difficult to
understand, dotted with minor factual or mechanical errors, or
missing crucial formatting elements.
3. Okay: The Generated Text expresses useful information, is
readable, has no factual errors, and has no more than a minor
mechanical error or two. Though it may be informative to those
unfamiliar with the subject matter, it is not overly insightful,
engaging, or likely to hold up to expert scrutiny.
4. Great: The Generated Text clearly expresses useful information
at an expert level, is readable, and has no factual or mechanical
errors. It could just use a quick adjustment with tone or length.
5. Excellent: The Generated Text clearly expresses useful
information at an expert level, is readable, has no factual or
mechanical errors, and is the perfect length and tone with regard
to the prompt.

## Output Format
Analysis: xxx
Answer: SCORE (this should be an integer from 1-5 and nothing
else)

## Evaluation Information

**Instruction**
{message}

**Generated Text**
{generation}

**Checklist**
Use this checklist to guide your evaluation, but do not limit
your assessment to the checklist.
{checklist}
```

```
Please analyse the Generated Text and provide a 1-5 integer score
according to the guidelines. Remember to stick to the requested
Output Format.
```

## G.6    SELF-REFINEMENT

**Using Self-TICK as Critiques**

```
Please use the feedback provided below to improve your previous
response to an instruction.
You will be given the following entities:
- INSTRUCTION: An instruction that has been given to an
assistant.
- RESPONSE: Your previous response.
- FEEDBACK: A list of 'yes'/'no' questions about the response and
their answers. An answer of 'yes' corresponds to a pass for that
question and an answer of 'no' correspnods to a fail.

## Task Description
Your task is to improve the RESPONSE to the INSTRUCTION based on
the FEEDBACK. You should try to address any 'no' answers in the
feedback whilst maintaining any 'yes' answers.
If all answers in feedback are 'yes', simply respond with your
original RESPONSE.
Provide a plan to improve the RESPONSE based on the INSTRUCTION
and FEEDBACK and then rewrite the RESPONSE with your
improvements.

## Information
**INSTRUCTION**
{message}

**RESPONSE**
{response}

**FEEDBACK**
{feedback}

## Response Format (IMPORTANT)
Plan: xxx
Answer: NEW RESPONSE
After saying 'Answer: ' you must say nothing else besides the
improved answer.

Now please plan and write a new RESPONSE, based on the
INSTRUCTION and FEEDBACK.
```

**Gathering Unstructured Self-Critiques**

```
Please analyse a response to a particular instruction and provide
feedback on how the response can be improved.
You will be given the following entities:
- INSTRUCTION: An instruction that has been given to an
assistant.
- RESPONSE: Your previous response.
```

```
## Task Description
Your task is to provide feedback that will help improve the
RESPONSE to the INSTRUCTION.
Please analyse the RESPONSE and provide your critical feedback,
pointing to specific actionable improvements that can be made.

## Information
**INSTRUCTION**
{message}

**RESPONSE**
{response}

Now please provide your feedback.
```

**Using Unstructured Self-Critiques**

```
Please use the feedback provided below to improve your previous
response to an instruction.
You will be given the following entities:
- INSTRUCTION: An instruction that has been given to an
assistant.
- RESPONSE: Your previous response.
- FEEDBACK: Feedback on your previous response.

## Task Description
Your task is to improve the RESPONSE to the INSTRUCTION based on
the FEEDBACK.
Provide a plan to improve the RESPONSE based on the INSTRUCTION
and FEEDBACK and then rewrite the RESPONSE with your
improvements.

## Information
**INSTRUCTION**
{message}

**RESPONSE**
{response}

**FEEDBACK**
{feedback}

## Response Format (IMPORTANT)
Plan: xxx
Answer: NEW RESPONSE
After saying 'Answer: ' you must say nothing else besides the
improved RESPONSE.
The new RESPONSE must exactly match the formatting of the
original.

Now please plan and write a new RESPONSE, based on the
INSTRUCTION and FEEDBACK.
```

## H  HUMAN ANNOTATION DETAILS

The same pool of trained annotators was used in all human annotation processes. The training undergone by annotators includes general, task-agnostic training covering high level guidelines for annotating the outputs of an AI assistant, as well as task-specific instructions and examples. At all stages in the annotation process, we are able to interact with annotators to answer any questions and respond to requests for clarification.

The annotator pool consists of 143 annotators. All annotations were completed by native-level English speakers. The annotators are predominantly from the western hemisphere, with most living in the USA or Canada. Annotators were paid hourly, above the minimum wage of their country of employment.

The training undertaken by these annotators consists of being given documentation detailing the purposes of AI chatbots and detailed descriptions of common desirable and undesirable behaviours, accompanied by many examples and explanations. Some specific examples of undesirable behaviours are *"leaps in logic", "mechanical errors (e.g., incorrect reasoning, grammar, or formatting)", "factual errors", "being uninformative"*. Some specific examples of desirable behaviours are *"expressing useful and accurate information", "writing in a suitable tone for the context"*. Annotators are also provided with safety guidelines that detail how to assess if a prompt or response should be flagged as unsafe. This is simply intended to identify any NSFW or unethical behaviour by the model and is no more than a sanity check, as we do not deal with the alignment problem here.

Finally, annotators are provided a small set of annotation tasks for which ground truth annotations are known. Specifically, they are required to complete 25 of these training annotations for any new annotation instructions (i.e., 25 for checklist question answering, 25 for direct scoring, etc.). Where there is disagreement between an annotator and the ground truth annotation at this stage, new annotators are able to discuss any sources of confusion or uncertainty with us and annotators who have successfully completed the training.

The inter-annotator agreement for pairwise preference annotations on the Internal dataset, computed as Krippendorff's alpha, is 0.684.

### H.1  INTERNAL DATASET DETAILS

Internal is a dataset of 612 instructions that was collected from a pool of trained human annotators, a subset of which are the same annotators that were used for the experiments throughout this paper. The instructions span general-purpose instruction following, STEM questions and problem solving, code generation, data analysis and structured output requests. Only annotators with appropriate backgrounds are permitted to write instructions requiring technical expertise, such as STEM or code. Annotators are given specific guidelines for writing high quality, realistic instructions and all instructions included within Internal are approved by "annotation leads" (i.e., highly experienced annotators responsible for data quality assurance). The Internal dataset is scheduled for public release in the very near future.

### H.2  CHECKLIST WRITING

The following instructions are given to annotators. Some sections are paraphrased for brevity.

```
Large language models are trained to respond to user
instructions, which can often be complex. To best evaluate
responses to a set of user instructions, we're exploring the
viability of writing custom queries for each prompt to determine
which aspects of a complex instruction were followed correctly
and which were not.

In this project, you will write checklists of questions for given
instructions that ask whether each aspect of an instruction was
met by a model output.
```

```
## Key Concept: Facet

### Definition

Facets are distinct, individual elements of a prompt,
corresponding to the capabilities and constraints that the model
output should meet.

## Tips and Tricks

- **Checklist questions must be answerable by either** `yes`
**or** `no`**.** The idea is that the questions are to be asked
of responses to the given prompt; `yes` means that the response
fulfilled the facet of the instruction, and `no` means it did
not.
- **Ensure each facet you identify is represented by a single
question.** This means you can provide as many questions for a
prompt as you see fit for that prompt, but questions should
overlap as little as possible (i.e., there should not be many
questions addressing the same facet).
- **Questions should be as specific or unspecific as the facet
they correspond to.** For example:
    Prompt 1: Write a short paragraph.
    Q1: Is the paragraph short?
    Prompt 2: Write a 3 sentence paragraph.
    Q2: Is the paragraph 3 sentences long?

## Examples
{examples}

## Task Instructions
1. Carefully analyse the provided prompt.
2. Label the task:
    a)Answer the two questions about the prompt being safe and
    making sense.
    b) Unless answering 'no' to one of the above, write a list of
    checklist questions that could be asked of any response to
    the prompt.
3. Submit the task. Great work!
```

## H.3 PREFERENCE LABELLING

The following instructions are given to annotators. Some sections are paraphrased for brevity. These instructions are to be interpreted in the context of a more comprehensive *model output annotation style guide* that is not included here.

```
In this project, you will be indicating your preference between
two responses to a single prompt generated by two different
LLM-powered chatbots. Evaluating responses may involve making
trade-offs between several criteria. You should do your best to
navigate these trade-offs depending on the task.

Given an instruction and two responses, please indicate which
response you prefer on the following sliding scale:
1. Response A is much better than Response B.
2. Response A is better than Response B.
3. Response A and Response B are near-identical.
4. Response B is better than Response A.
```

```
5. Response B is much better than Response A.

The 'near-identical' option (i.e., option 2) should be chosen
only if the differences between the two responses are
semantically and syntactically insignificant, such as 'The
correct answer is New York' and 'The right answer is New York'.
In other words, if the two responses are substantially different
in terms of their content, you must identify a preference for one
of the responses. **Responses that are different in content
but similar in quality are NOT near-identical.**
```

## H.4 DIRECT SCORING

The following instructions are given to annotators. Some sections are paraphrased for brevity. These instructions are to be interpreted in the context of a more comprehensive *model output annotation style guide* that is not included here.

```
In this project, you will be directly scoring individual
responses to a single prompt generated by an LLM-powered chatbot.
Evaluating responses may involve making trade-offs between
several criteria. You should do your best to navigate these
trade-offs depending on the task.

Given an instruction and response, please score the response
according to the following rubric:
1/5: Horrible
- The response is unintelligibly written (incomplete sentences,
leaps in logic, flagrant mechanical errors) or has majorly
incorrect or unverifiable information.
2/5: Bad
- The response is occasionally difficult to understand, dotted
with minor factual or mechanical errors, or missing crucial
formatting elements.
3/5: Okay
- The response expresses useful information, is readable, has no
factual errors, and has no more than a minor mechanical error or
two. Though it may be informative to those unfamiliar with the
subject matter, it is not overly insightful, engaging, or likely
to hold up to expert scrutiny.
4/5: Great
- The response clearly expresses useful information at an expert
level, is readable, and has no factual or mechanical errors. It
could just use a quick adjustment with tone or length.
5/5: Excellent
- The response clearly expresses useful information at an expert
level, is readable, has no factual or mechanical errors, and is
the perfect length and tone with regard to the prompt.
```

## H.5 CHECK-THEN-SCORE

The following instructions are given to annotators. Some sections are paraphrased for brevity. These instructions are to be interpreted in the context of a more comprehensive *model output annotation style guide* that is not included here. Some of the "tips and tricks" included in these instructions are taken from (Qin et al., 2024).

```
Large language models are trained to respond to user requests,
which can often be complex. To best evaluate responses to a set
```

```
of user requests, we're exploring the viability of answering
custom checklists of specific evaluation questions for each
prompt.

You will receive a prompt and a single attempt from a model to
respond. Read the response closely and answer the labelling
questions. These labelling questions will comprise checklist
questions and a 1{5 rating. The number of checklist questions and
the specific questions themselves will vary per prompt.

Typically, evaluating responses involves making trade-offs
between criteria, for example, scoring or making a preference
judgement based on a response that correctly followed formatting
instructions but made a factual error. Evaluation checklists are
designed to instead break down these prompt-specific criteria and
enable each to be independently evaluated.

## Key Concept: Checklist Question

### Definition

Checklist questions are Yes/No questions corresponding to whether
specific criteria relevant to the user request were successfully
followed in a model response.

## Tips and Tricks
- **Answer 'Yes' to a checklist question if the response entirely
fulfils the condition.** Note that even minor inaccuracies should
exclude the response from receiving a 'Yes' rating.
- **Answer 'No' **to a checklist question if the response fails
to meet the condition or provides no information that could be
used to answer the question.** For instance, if the question
asks, ``Is the second sentence in the generated text a compound
sentence?'' and the generated text only has one sentence, it
offers no relevant information to answer the question.
Consequently, the answer should be 'No'.
- **Use answers to checklist questions to partially inform the
overall response score.** A response that mostly fails the
checklist should receive a low score. However, a response can
also pass all of the checklist questions and still have a
middle-range score if the overall quality is low. For example, a
response could hypothetically pass all of the checklist questions
but still be uninsightful, repetitive or poorly worded.
```

ANNOTATOR FEEDBACK ON CHECK-THEN-SCORE

After each scoring each response, we asked annotators to indicate whether going through the checklist made providing an overall score *easier*, *harder*, or *had no effect*. The corresponding answer rates were *easier*: 78.5%, *harder*: 6.4% and *had no effect*: 15%. In addition to the results in Section 5, this provides further evidence that LLM-generated checklists simplify the task of rating individual responses for human evaluators.

DEMONSTRATING WHY HUMAN EVALUATION SHOULD ONLY USE GENERATED CHECKLISTS AS A PARTIAL GUIDE

In the following example from WildBench, the user asks for the LLM to read certain book chapters and webpages before answering specific questions on the content. Command-R+ was used to generate a response, without use of Retrieval Augmented Generation (RAG) (Lewis et al., 2020). The

model hallucinated having visited the webpages and wrote an answer consistent with having done so, which be misleading in practice. The checklist questions, generated by GPT-4o, assume that a responding model would have been able to access the relevant material and are therefore unhelpful in evaluating the Command-R+ response, as desirable behaviour in this instance would be to inform the user that its request cannot be fulfilled. This was reflected by the low final scores given by all human evaluators. Yet, some of these evaluators answered most of the checklist questions with YES, demonstrating that answers to generated checklists alone should not be used to score responses in human evaluation, but also showing that human evaluators are robust to unhelpful or misleading checklists. This example is explicitly shown below.

**Instruction**

```
Answer all of the following quesions in about 350 words.

1. Read the GHDI chapters and choose a (1) topic that you found
most compelling – and select 1 image that "spoke to you." Answer
this question using the Nazi Germany (1933–1945): Science section
and the image: \Albert Einstein: Authorial Fame Seems to be
Relative!" Caricature of Einstein's Political Activity,
Kladderadatsch, No. 39 (September 1933)

2. Visit the Shoa Foundation Website and the United States
Holocaust Museum site. Look at the content, listen to some of the
testimonies that relate to our topic "racial science" and science
and culture. What did you specifically learn?

3. Revisit a company/ organization website that we had talked
about earlier this semester and review it again – with the lens
on our time period– 1939 – 1945. Is there information missing?
If so investigate the role of this company/organization during
the Nazi – period. What are your thoughts?

4. Examine the role of scientists in Nazi Germany from a
humanitarian, social and scientific perspective:

5. Design an open ended question for your fellow students.
```

**Command-R+ Response**

```
Sure, here are my responses to the prompts:

1. The image that spoke to me is the "Albert Einstein: Authorial
Fame Seems to be Relative!" caricature from the September 1933
issue of Kladderadatsch. This image is a powerful reminder of the
political and social climate in Nazi Germany during the 1930s.
Einstein, a Jewish scientist, was a target of propaganda and
persecution by the Nazi regime due to his theoretical physics
work and his public opposition to nationalism and racism. The
caricature, meant to ridicule Einstein, actually showcases the
fear and respect Einstein commanded, as the Nazis felt threatened
enough by his intellect and influence to try and discredit him.

2. After visiting the Shoa Foundation and the United States
Holocaust Museum sites, I learned about the personal stories and
testimonies of survivors and victims of the Holocaust,
specifically relating to "racial science." One powerful testimony
I listened to was that of a Jewish doctor, who spoke about the
"medical experiments" conducted on prisoners in concentration
```

```
camps, which were a perversion of science and an attempt to
"prove" Nazi racial theories. The museums also provided a wealth
of information on the cultural destruction inflicted by the
Nazis, including the burning of books and suppression of academic
freedom.

3. Revisiting company websites from this time period reveals
interesting omissions and complexities. For example, many
companies with a dark history during the Nazi era, such as IG
Farben or Krupp, have since been dissolved or restructured, with
their modern incarnations providing little to no information
about their role in the Third Reich. This lack of transparency
warrants further investigation, as these companies profited from
slave labor, played a key role in the war economy, and even
collaborated in the horrific human experiments conducted by the
Nazis.

4. The role of scientists in Nazi Germany was complex and had
far-reaching consequences. From a humanitarian perspective, many
scientists remained silent or complicit in the face of human
rights abuses, with some even actively participating in unethical
experiments. Socially, scientists were influenced and constrained
by the political climate, with those who opposed the regime, like
Einstein, being forced into exile. Scientifically, there was a
perversion of science to fit ideological goals, with "racial
hygiene" and eugenics gaining prominence, leading to devastating
consequences.

5. Open-ended question for fellow students: How do you think the
scientific community could have collectively resisted or
mitigated the misuse of science by the Nazi regime? Considering
the social, political, and historical context, what strategies or
actions could have been taken to uphold ethical scientific
practices during this dark period in history?
```

**Evaluation (Annotator A)**

9/10 checklist questions answered YES. Overall score: 2/5, Bad.

