# OpenReview forum: "TICKing All the Boxes: Generated Checklists Improve LLM Evaluation and Generation"
_ICLR.cc/2025/Conference — Submitted to ICLR 2025_

### Official Review · Reviewer_sSrg · 2024-10-23

**Soundness:** 3
**Presentation:** 3
**Contribution:** 3
**Rating:** 6
**Confidence:** 5

**Summary:**

This paper explores developing an automated evaluation benchmark to assess the instruction-following ability of large language models. Their study is based on the idea that asking LLMs to evaluate response qualities with a set of detailed requirements provides more reliable assessments than asking LLMs to provide a holistic evaluation directly, as proposed by InfoBench. The major finding of this paper is that LLMs can also prepare the decomposed questions (i.e., the checklist) for arbitrary user prompts, scaling up this framework to the next level of automation. Also, they find that the LLM-generated checklist could further help LLMs to provide self-refined responses.

**Strengths:**

1. This paper removes the major constraint of manually constructing checklists of prior works, significantly improving the scalability of automated instruction-following benchmarks.
2. It is interesting that the checklist can help LLMs refine their initial responses.
3. The paper is well-written and well-organized.

**Weaknesses:**

1. The metrics to evaluate the similarities between the human-crafted and LLM-generated checklists can be improved. In particular, those lexical-matching metrics (i.e., BLEU and ROUGE) should be replaced with more semantic ones. For example, [1] evaluates the quality of LLM-generated rubrics versus to human-crafted ones with BERTScore. Further reporting the percentage of recalled human-crafted check items and the percentage of precise LLM-generated check items will be better.

2. This paper fails to discourse the details of human study. In this paper, many experiments are conducted with human annotators. The authors should discuss some basic information about the annotations, such as the statistics of their demographic information, the training procedures for the annotators, and the internal agreement among the annotators.

[1] Unveiling Scoring Processes: Dissecting the Differences between LLMs and Human Graders in Automatic Scoring.

**Questions:**

Please see the suggestions in Weaknesses.

---

> ### Author Response · Authors · 2024-11-13
> **Taking on suggestions**
>
> We would like to thank the reviewer for their precise and informative review. We are glad that the reviewer sees our paper as “significantly improving the scalability of automated instruction-following benchmarks” and finds it “interesting that the checklist can help LLMs refine their initial responses”.
>
> We directly address the weaknesses raised by the reviewer in the updated manuscript, as described below.
>
> > Lexical-matching metrics should be replaced with more semantic ones [such as] BERTScore.
>
> We thank the reviewer for suggesting this and have included BERTScore as a column in Table 2 of the updated manuscript. These results further demonstrate the similarity between LLM-generated and gold standard human-written checklists, thus strengthening this part of the paper. We acknowledge that reporting the percentage of recalled gold standard checklist items would be another informative metric, however doing so would require further costly human annotations, as judging whether items from two different checklists are precisely the same is an ambiguous task. However, we hope that the addition of BERTScore in combination with reporting the count MAE provides sufficient evidence that the checklists are similar in terms of count and content. Finally, we would also like to point the reviewer to Table 3 (a), which shows that the downstream evaluation scores from using gold standard or LLM-generated checklists are highly correlated, demonstrating that the checklists also lead to similar evaluations on aggregate.
>
> > The paper fails to discourse the details of human study.
>
> We would again like to thank the reviewer for identifying this shortcoming. We have updated the manuscript to include further details on the training of annotators and report the inter-annotator agreement. This information is now at the top of Appendix H.
>
> We are confident that these changes strengthen our paper and hope that the reviewer finds that their suggestions have been directly addressed.

---

> ### Comment · Reviewer_sSrg · 2024-11-20
> **Comments to Author Response**
>
> My first concern has been addressed. However, I still look forward to seeing the precision/recall metrics for such a rule-decomposition task.
>
> My second concern has not been addressed. In particular, it provides some general information about human study without clarifying any basic numbers, like the total number of annotators. In addition, I am very confused about one statement in their attached content: "... if a prompt or response should be flagged as unsafe." Since the whole paper has not discussion about the safety alignment problem, where is the "unsafe" coming from? Combining these two factors, I suspect that the attached human study details in Appendix H are written by an LLM, which cannot discourse the real procedure of the human study. I will turn down my confidence because of this concern.

---

> ### Author Response · Authors · 2024-11-21
> **Responding to further comments**
>
> We would like to thank the reviewer for their deeply engaging with our paper during this discussion period. We are pleased to take the opportunity to address the reviewer's remaining concerns.
>
> > I still look forward to seeing the precision/recall metrics for such a rule-decomposition task.
>
> Given that there is subjectivity in assessing whether or not two items from two different checklists are the same, we have made an effort to deliver on this request by using an LLM (GPT-4o) to label matching items between a model generated and human-written checklist. From this we can compute precision, recall and F1 with respect to the gold standard human-written checklists.
>
> This analysis gave us the following results:
>
> Command-R+ - Precision: 0.766, Recall: 0.702, F1: 0.712
>
> GPT-4o - Precision: 0.811, Recall: 0.781, F1: 0.773
>
> Given that we are unable to use human annotation for this analysis within the discussion period, we hope that this approach of using a strong LLM to label equivalent checklist items addresses the reviewer's request.
>
> > It provides some general information about human study without clarifying any basic numbers, like the total number of annotators. In addition, I am very confused about one statement in their attached content: "... if a prompt or response should be flagged as unsafe." Since the whole paper has not discussion about the safety alignment problem, where is the "unsafe" coming from? Combining these two factors, I suspect that the attached human study details in Appendix H are written by an LLM, which cannot discourse the real procedure of the human study. I will turn down my confidence because of this concern.
>
> **We assure the reviewer that the additions to Appendix H were not written by an LLM.** We have included a number of expansions on the points that were added to that appendix in a new revision of the manuscript in the hopes that this clears up *specifically* what was meant by each point. We also include the revised content here:
>
> The annotator pool consists of 143 annotators. All annotations were completed by native-level English speakers. The annotators are predominantly from the western hemisphere, with most living in the USA or Canada. Annotators were paid hourly, above the minimum wage of their country of employment.
>
> The training undertaken by these annotators consists of being given documentation detailing the purposes of AI chatbots and detailed descriptions of common desirable and undesirable behaviours, accompanied by many examples and explanations. Some specific examples of undesirable behaviours are "leaps in logic'', "mechanical errors (e.g., incorrect reasoning, grammar, or formatting)", "factual errors", "being uninformative"}. Some specific examples of desirable behaviours are "expressing useful and accurate information", "writing in a suitable tone for the context"}. Annotators are also provided with safety guidelines that detail how to assess if a prompt or response should be flagged as unsafe. This is simply intended to identify any NSFW or unethical behaviour by the model and is no more than a sanity check, as we do not deal with the alignment problem here.
>
> Finally, annotators are provided a small set of annotation tasks for which ground truth annotations are known. Specifically, they are required to complete 25 of these training annotations for any new annotation instructions (i.e., 25 for checklist question answering, 25 for direct scoring, etc.).  Where there is disagreement between an annotator and the ground truth annotation at this stage, new annotators are able to discuss any sources of confusion or uncertainty with us and annotators who have successfully completed the training.
>
> The inter-annotator agreement for pairwise preference annotations on the Internal dataset, computed as Krippendorff's alpha, is 0.684.
>
> **We sincerely hope that this addresses the reviewer's remaining concerns**, especially regarding the accusation of using LLM generated content, which we take very seriously.

---

> ### Comment · Reviewer_sSrg · 2024-11-21
> **Comments to Author Response**
>
> I am happy to see these discussions and my concerns have been addressed. Please include them in your manuscript. I will turn back to my initial evaluation scores.

---

### Official Review · Reviewer_vpZ1 · 2024-11-01

**Soundness:** 2
**Presentation:** 3
**Contribution:** 2
**Rating:** 5
**Confidence:** 4

**Summary:**

To evaluate the instruction-following capabilities of large language models (LLMs), this paper introduces a method called TICK (Targeted Instruct-evaluation with ChecKlists). TICK leverages the in-context learning abilities of LLM to break down complex instructions into a series of yes/no questions, forming a checklist. The LLM is then used to score this checklist. Initially, the paper demonstrates the advantages of the TICK assessment method through extensive human consistency experiments. Subsequently, the effectiveness of the TICK method is validated through experiments involving self-refinement, Best-of-N selection, and assistance with human annotations.

**Strengths:**

1. TICK enhances the transparency and interpretability of the evaluation process by breaking down the assessment task into a series of specific YES/NO questions. This fine-grained evaluation approach helps to more accurately identify the strengths and weaknesses in the model's output.

2. This paper conducts extensive automated and manual consistency experiments to quantify and demonstrate the advantages of the TICK evaluation method.

**Weaknesses:**

1. The core of the proposed method in this paper lies in using in-context learning to break down instructions into a checklist for self-validation and refinement, as well as for best-of-N selection. However, employing decomposed checklists for instruction evaluation ,validation and refinement is not new, as seen in work like FollowBench, InfoBench, and Self-Contrast. The fundamental differences and substantive contributions of this work compared to existing approaches, particularly in terms of evaluation methods and self-improvement strategies, need to be more clearly defined.

2. There is a lack of in-depth discussion regarding the efficiency of the proposed evaluation method.

**Questions:**

1. Although checklists introduce a certain level of structure, they typically only express parallel relationships. When the content to be verified involves more complex logical relationships, such as selective, chain relationships, or their combinations (for example, tasks in ComplexBench), how can the effectiveness of checklists be ensured?

2. A notable feature of instruction-following tasks is that verification points are directly reflected in the instructions (such as text style, word count limits, etc.), making it relatively easy to break down the task into different verification points and generate checklists. However, for a wider range of task types, especially in fields involving symbolic reasoning like mathematics and programming, how can the application methods and advantages of checklists be demonstrated?

3. For models with different capability levels, particularly some weaker or smaller-scale language models (LLMs), how do they perform in terms of decomposing checklists and accurately scoring?

---

> ### Author Response · Authors · 2024-11-13
> **Addressing concerns**
>
> We are pleased that the reviewer finds that the paper “conducts extensive automated and manual consistency experiments to quantify and demonstrate the advantages of the TICK evaluation method”. However, we find it a shame that the reviewer has not acknowledged Section 4 of the paper, beyond claiming that using checklists for refinement is not new, despite the fact that no prior work on using checklists for in-context self-improvement exists to the best of our knowledge. Our paper demonstrates that Self-TICK (STICK) significantly improves in-context self-refinement, an increasingly common practice for improving response quality by expending more compute at inference time [1]. Table 1 demonstrates that end-to-end checklist self-evaluations enable purely in-context self-correction in challenging code, reasoning and mathematics problems, despite the reviewer’s claims that we do not consider these task types. We are therefore led to believe that the reviewer has not considered these results in their assessment and would like to highlight their significance, especially in light of several recent works suggesting that purely in-context self-correction is yet to be demonstrated in these settings [2, 3].
>
> > Employing decomposed checklists for instruction evaluation, validation and refinement is not new, as seen in work like FollowBench, InFoBench, and Self-Contrast.
>
> FollowBench and InFoBench use expensive-to-gather, human-written checklists, which limits the use of checklist-based evaluations to their predefined prompt datasets, whereas TICK is substantially cheaper and generally applicable (as acknowledged by Reviewer sSrg), which is what enables Section 4 of the paper, where an LLM can perform checklist-based self-evaluations on-the-fly. Self-Contrast is not closely related to our paper, being a similarity-based method for alignment involving two fine-tuning phases. To the best of our knowledge, no prior work uses decomposed evaluation structure to enable improved iterative self-refinement and even self-correction, which recent work has suggested requires RL fine-tuning to achieve [2].
>
> > There is a lack of in-depth discussion regarding the efficiency of the proposed method.
>
> We thank the reviewer for identifying this. Scaling inference compute by sampling more or longer generations is an increasingly common practice for improving LLM capabilities on problems that are otherwise challenging [1, 4, 5]. We therefore see the improvements of TICK and STICK as being due to an effective way of improving evaluation and refinement quality in exchange for additional inference costs. To further address this concern, we additionally compare to the most common approach to inference scaling of majority vote among K generations sampled in parallel (i.e., Maj@K) [5] in Table 4 of the updated manuscript. We do this for preference-based LLM-as-judge evaluation and direct scoring with K=32 and still using Chain-of-Thought for the evaluator in each case. The results demonstrate that this improves both LLM-as-judge preferences and direct scores, but that both still perform worse than TICK, highlighting that TICK makes more efficient use of additional tokens than majority vote.
>
> ## Answering questions
>
> 1. We acknowledge that checklists do not capture sequential dependencies by default and see the automatic construction of evaluation rubrics for agentic tasks as an exciting direction for future work. Whilst ComplexBench explicitly constructs a dataset of instructions with constraint dependencies and has human annotators write checklists that reflect this, simply prompting the LLM to “opt for ’NO’ if the generated text provides no information that could be utilised to answer the question” implicitly captures the fact that a checklist question should be answered ‘NO’ if a question higher up a dependency chain is answered ‘NO’, as is done in this work and InFoBench.
>
> 2. We explicitly prompt the LLM to include “implicit criteria that are generally important for an instruction’s problem domain” in the checklist (line 174 in the manuscript), the positive effect of which can be observed in the examples of generated checklists in the appendix and in the positive STICK results on precisely the fields mentioned by the reviewer (Table 1 of the manuscript).
>
> 3. We have included results for Llama-3.1-8B-Instruct in Table 2 and Table 3 (b) to address this. We see that it performs only marginally worse than larger models at both generating and answering checklist questions.
>
> [1] Snell et al, Scaling LLM Test-Time Compute Optimally can be More Effective than Scaling Model Parameters, 2024
>
> [2] Kumar et al, Training Language Models to Self-Correct via Reinforcement Learning, 2024
>
> [3] J. Huang et al, Large Language Models Cannot Self-Correct Reasoning Yet, 2023
>
> [4] Madaan et al, Self-Refine: Iterative Refinement with Self-Feedback, 2023
>
> [5] X. Wang et al, Self-Consistency Improves Chain of Thought Reasoning in Language Models, 2022

---

> > ### Author Response · Authors · 2024-11-25
> > **Request for engagement**
> >
> > Dear reviewer,
> >
> > As the discussion period draws to a close, we ask if you would consider our response to your initial review so that we can act on any unaddressed concerns and further improve the paper.
> >
> > With kind regards,
> >
> > The Authors

---

> > ### Comment · Reviewer_vpZ1 · 2024-11-25
> > **Response to authors**
> >
> > Thank you for the reviewer's response. I am willing to increase my score to 5 points.
> >
> > Regarding question 2, I still remain confused about the scope of application for the checklist.
> >
> > For example, in a multi-step reasoning process or a forward-solving mathematical problem, it would be a significant challenge for the model to directly generate a verification list from the problem.
> >
> > Here's a random example:
> > A rectangular swimming pool's length is 1.5 times its width. If both the length and width are increased by 2 meters, the area will increase by 64 square meters. Find the original perimeter of the swimming pool.
> > What would the checklist look like for this problem?
> >
> > In the mathematics and code cases exemplified by the author in the response to reviewer KpU7, I still feel that this checklist is not in-depth and only seeks information on the surface level.

---

> > > ### Author Response · Authors · 2024-11-25
> > > **Further discussion**
> > >
> > > Thank-you for your response and for increasing your score.
> > >
> > > To further address your question regarding the scope and detail of checklists for reasoning problems, we would like to clarify that the (self-)evaluation checklists work *better* than existing approaches to critiquing a response (i.e., unstructured critiquing, relying on fixed, human-written checklists, or providing a black-box evaluation such as a score or preference), but do not claim that they are able to extract *all* information relevant to evaluation, which is of course a grand challenge for research in this direction.
> > >
> > > However, we do believe that the self-correction results in Table 1, facilitated by checklist-based self-critiques on reasoning-intensive tasks, are striking given that prior work indicates that in-context self-correction fails in these settings without RL fine-tuning [1] or human assistance [2].
> > >
> > > **New result**: To further address this point of uncertainty for the reviewer, we have run the self-correction on experiment for Command-R+ on LiveBench (Mathematics) (Table 1) where checklists are generated by conditioning on both the instruction *and response*. We thank the reviewer for raising this point, as we observe even stronger self-improvement (+1.8 -> +2.4) and that the generated checklists are indeed more specific (see the example for your example mathematical reasoning question below). We humbly ask if this clarification and additional result is sufficient for you to consider further increasing your score. We will endeavour to run this additional experiment for all of Table 1 in time for a potential camera-ready deadline. Note that we cannot use response conditioning for TICK or Best-of-N STICK, as this would not be a fair way to evaluate multiple responses to the same instruction.
> > >
> > > **Your example question**:
> > >
> > > **GPT-4o generated checklist (only instruction conditioned)**:
> > >
> > > Does the response correctly interpret that the length is 1.5 times the width?
> > >
> > > Does the response set up a correct equation for the change in area based on the increased dimensions (area increase = 64 square meters)?
> > >
> > > Does the response correctly solve the equations to find the original width and length of the swimming pool?
> > >
> > > Does the response calculate the original perimeter using the correct formula P = 2 × (length + width)?
> > >
> > > Is the mathematical reasoning and calculation in the response free of errors?
> > >
> > > Does the response provide the final answer clearly and explicitly state the original perimeter?
> > >
> > > **GPT-4o generated checklist (instruction and response conditioned)**:
> > >
> > > Does the response clearly define variables (e.g., width w) and explain the relationship between the width and length (length = 1.5w)?
> > >
> > > Does the response correctly identify that the change in area after increasing dimensions is 64 square meters?
> > >
> > > Does the response set up the relationship between the original dimensions and the new dimensions accurately?
> > >
> > > Is the original area expressed correctly?
> > >
> > > Is the new area expressed correctly?
> > >
> > > Does the response correctly set up the equation for the difference in areas as new area - original area = 64?
> > >
> > > Does the response expand and simplify the expression for the new area correctly without algebraic errors?
> > >
> > > Is the perimeter correctly calculated using the formula P = 2 x (length + width)?
> > >
> > > [1] Kumar et al, Training Language Models to Self-Correct via Reinforcement Learning, 2024
> > >
> > > [2] Huang et al, Large Language Models Cannot Self-Correct Reasoning Yet, 2023

---

### Official Review · Reviewer_KpU7 · 2024-11-03

**Soundness:** 3
**Presentation:** 4
**Contribution:** 2
**Rating:** 6
**Confidence:** 4

**Summary:**

The paper aims to measure and enhance LLM performance in instruction-following tasks by leveraging a powerful model to generate checklists based on the given instructions.
The key contributions include:
1. Proposing a prompt to generate checklists for each instruction.
2. Validating the high similarity between checklists generated by advanced LLMs and those created by humans across several benchmarks.
3. Showing that the judge score derived from aggregating checklists yields a pass ratio that closely aligns with human scores, highlighting the potential of using the checklist to improve the performance of LLM-as-judge.
4. Showcasing that self-refinement guided by the generated checklists leads to higher performance improvements compared to unstructured feedback.
5. Allowing human annotators to reference the model-generated checklists results in enhanced inter-annotator agreement.

**Strengths:**

Originality: This paper analyzes the quality of checklists generated by advanced LLMs and how they can be used to improve LLM-as-judge and high-quality instruction selection. It can provide experiment results for practitioners who want to use these checklists to enhance the performance of LLMs as judges, offering valuable insights.

Quality: The overall experimental analysis is thorough, including validation of LLM-generated checklists to human-generated checklists. It also features corresponding analyses on the use of checklists for self-refinement and their application as the reference for human annotators.

Clarity: The paper is written clearly, making it easy to follow and understand.

Significance: The topic of LLMs as judges is highly relevant, and the findings of this study may offer significant insights for the industry.

**Weaknesses:**

Novelty: Given multiple works on using checklists to enhance the performance of LLMs as judges, this paper’s contribution lies in enabling LLMs to generate their own checklists and validating their feasibility. The approach involves introducing a specific prompt to elicit the checklist from the LLM. However, this requires the LLM to first follow a complex set of instructions to generate the checklist, which places even higher demands on the model’s capabilities than the instruction-following task itself.

Experimental Limitations: From an experimental perspective, the study could benefit from considering a wider range and a larger scale dataset. Currently, it only examines three benchmarks: Internal, InfoBench, and WildBench.

Expense: The existing design is computationally expense during inference time since it requires a large number of tokens and multiple generations during the self-refinement stages. How to distill this ability or reduce this expense can be a good direction.

**Questions:**

1. For table 2, why don't you consider the semantic similarity metrics such as scores generated by natural language inference models? BLEU and Rouge style metrics sometimes can be unreliable.

---

> ### Author Response · Authors · 2024-11-13
> **Taking on comments and providing clarifications**
>
> We thank the reviewer for their detailed review. We are pleased that the reviewer sees our paper as “offering valuable insights” and that the “overall experimental analysis is thorough”. We strongly believe that the weaknesses raised by the reviewer are addressed by the following clarifications.
>
> > [The approach] requires the LLM to first follow a complex set of instructions to generate the checklist.
>
> Our results demonstrate that current LLMs are already capable of generating checklists that are similar to gold standard human-written checklists (Table 2 and Table 3 (a)), including smaller, open-source models, such as Llama-3.1-8B for which results have been added in the updated manuscript. Additionally, checklist self-feedback (i.e., STICK) proves effective at enabling self-correction where unstructured self-feedback fails (Table 1), demonstrating that checklist generation and answering in fact *eases* the problem of answering the original instruction and thus cannot be more difficult than answering the original instruction.
>
> > It only examines three benchmarks: Internal, InFoBench, and WildBench.
>
> As shown in Table 1, we also evaluate on LiveBench, which spans a range of task categories covering reasoning, mathematics, code, language and more. Additionally, both Internal and WildBench cover a very broad spectrum of instructions, with WildBench instructions being taken from a wide range of real-world interactions with chatbots. We believe that the four benchmarks considered cumulatively provide strong evidence for the benefits of using automatically generated checklists to structure automatic evaluation.
>
> > The existing design is computationally expensive during inference time.
>
> Scaling inference compute as an alternative to scaling training compute has emerged as an exciting paradigm for further improving LLM capabilities [1, 2], with self-refinement [3] and self-correction [4, 5] becoming popular research directions. We convincingly demonstrate that checklist-based self-evaluations are an effective way obtaining greater benefits from increased inference compute, whether by iterative refinement (Table 1 & Figure 3), or Best-of-N selection (Table 5). As a further investigation of how TICK compares to alternative approaches to assigning more inference compute to the task of evaluation, we have added a comparison to majority vote among 32 parallel sampled evaluations (i.e., Maj@32) for preference and direct scoring in Table 4 of the updated manuscript. We see that doing so improves agreement between the subsequent preferences or scores and human evaluations, but that they remain worse than TICK.
>
> ## Answering questions
>
> 1. We thank the reviewer for this suggestion. We have included results using the semantic similarity metric BERTScore in Table 2 of the updated manuscript.
>
> [1] Snell et al, Scaling LLM Test-Time Compute Optimally can be More Effective than Scaling Model Parameters, 2024
>
> [2] OpenAI, o1, 2024
>
> [3] Madaan et al, Self-Refine: Iterative Refinement with Self-Feedback, 2023
>
> [4] Kumar et al, Training Language Models to Self-Correct via Reinforcement Learning, 2024
>
> [5] Gou et al, CRITIC: Large Language Models Can Self-Correct with Tool-Interactive Critiquing, 2024

---

> > ### Comment · Reviewer_KpU7 · 2024-11-22
> > **Thank you for the response**
> >
> > I have reviewed the author’s reply, as well as the other reviewers' comments and the corresponding responses. Overall, I am happy to see that the checklists generated by the LLM are as reliable as human-written checklists, and that these can be used to iteratively enhance the model’s performance. So I am willing to increase the score to 7.
> >
> > Additionally, I found another reviewer’s question quite interesting: "A notable feature of instruction-following tasks is that verification points are directly reflected in the instructions (such as text style, word count limits, etc.), making it relatively easy to break down the task into distinct verification points and generate checklists. However, for a broader range of task types, especially in domains involving symbolic reasoning such as mathematics and programming, how can the application and advantages of checklists be demonstrated?" Could you provide examples to illustrate the positive impact of the generated checklists in such scenarios?

---

> > > ### Author Response · Authors · 2024-11-22
> > > **Thank you for continued discussion**
> > >
> > > Thank you for engaging with our response and helping to improve our paper. We are pleased that you are happy with our results and appreciate your willingness to increase your score.
> > >
> > > We would also like to thank you for engaging deeply with our paper by drawing interesting questions from the other reviews and are more than happy to answer your question. Enhancing the model's performance on LiveBench (results in Table 1 of the manuscript), which covers "Coding", "Data Analysis", "Mathematics" and "Reasoning" tasks, is enabled by our checklist-based approach, as Table 1 reveals that non-checklist based self-critiques cause performance to drop in these domains involving symbolic reasoning. [1, 2] argue that standard approaches to self-refinement wrongly identify errors in the LLM's previous response. When we inspect generated checklists for tasks like coding problems, we can see how using checklists for self-evaluation can get around this issue by grounding the self-correction process in highly specific aspects of a desirable response.
> > >
> > > We include an example prompt and generated checklist, which is in the appendix of the manuscript, for a LiveBench Coding task, followed by a Math task below:
> > >
> > > **Instruction:** You are given a 0-indexed integer array nums and an integer k. You can perform the following operations on the array at most k times: Choose any index i from the array and increase or decrease nums[i] by 1. The score of the final array is the frequency of the most frequent element in the array. Return the maximum score you can achieve. The frequency of an element is the number of occurrences of that element in the array. Only write the missing portion of the code, not the entire code.
> > >
> > > Constraints:
> > >
> > > 1 <= nums.length <= 10^5
> > >
> > > 1 <= nums[i] <= 10^9
> > >
> > > 0 <= k <= 10^14
> > >
> > > **GPT-4o generated checklist:**
> > >
> > > - Does the response only include the missing portion of the code and nothing else?
> > >
> > > - Does the response correctly continue from the given starting code?
> > >
> > > - Does the response handle the operations correctly to modify elements at most 'k' times to maximize the frequency of the most frequent element?
> > >
> > > - Does the response correctly implement logic to track and calculate the frequency of the most frequent element in the array?
> > >
> > > - Does the response ensure the final implementation is syntactically correct and free form errors?
> > >
> > > - Is the approach efficient given the constraints of the problem ('1 <= nums.length <= 10^5', '1 <= nums[i] <= 10^9', '0 <= k <= 10^14')?
> > >
> > > And for a math problem:
> > >
> > > **Instruction:** Differentiate the following function: sin(7x^4 + 4)cos(9-x). Please put your final answer in a [].
> > >
> > > **GPT-4o generated checklist:**
> > >
> > > - Does the response correctly apply the product rule to differentiate the given function?
> > >
> > > - Does the response correctly differentiate the individual components sin(7x^4 + 4) and cos(9 - x)?
> > >
> > > - Are the intermediate steps clear and logically presented?
> > >
> > > - Is the final answer correctly boxed using the [] notation?
> > >
> > > We hope that this answers your question and thank you again for your engagement and willingness to increase your score.
> > >
> > > [1] Kumar et al, Training Language Models to Self-Correct via Reinforcement Learning, 2024
> > >
> > > [2] Huang et al, Large Language Models Cannot Self-Correct Reasoning Yet, 2023

---

> > > > ### Author Response · Authors · 2024-11-25
> > > > **Further to discussion**
> > > >
> > > > Dear reviewer,
> > > >
> > > > Thank you again for providing an initial review that prompted us to produce additional experimental results that have improved the paper and for engaging in in-depth discussion about our work. We would to check in that our previous responses addresses your previous question, as we see that your score remains at 6.
> > > >
> > > > With kind regards,
> > > > The Authors

---

> > > > > ### Author Response · Authors · 2024-12-03
> > > > > **Final request for engagement**
> > > > >
> > > > > Dear reviewer,
> > > > >
> > > > > We'd once more like to thank you for your prior review and follow-up engagement with our paper. We would like to clarify that our previous comment answered your question and confirm that you are still willing to raise your score, as it has not yet been updated.
> > > > >
> > > > > Many thanks,
> > > > >
> > > > > The Authors

---

### Official Review · Reviewer_saAj · 2024-11-04

**Soundness:** 2
**Presentation:** 3
**Contribution:** 3
**Rating:** 5
**Confidence:** 4

**Summary:**

The authors propose TICK, a method that uses LLMs to decompose instructions into checklists composed of several YES/NO choices to address limitations in standard evaluation metrics like Elo rating and direct scoring. This approach provides a more interpretable evaluation by breaking down instructions into specific criteria. They further introduce STICK, which refines LLM responses using self-assessment based on these checklists, achieving substantial improvements compared to traditional refinement methods. Experiments demonstrate that using LLMs for checklist generation is feasible and reliable. Also, using checklists for evaluation aligns with human annotations. Based on TICK, STICK enhances the quality of LLM outputs beyond vanilla-refinement approaches. Additionally, the authors find that using checklists in human annotation significantly increases inter-annotator agreement, making the evaluation process more consistent and reliable.

**Strengths:**

- The automatic evaluation method using LLMs as judges is novel and significant. The authors present an effective and interpretable protocol for evaluating and refining generated text.
- Comprehensive experiments and detailed analyses are provided to support the effectiveness of the proposed methods.
- The paper is well-written and easy to follow, making it accessible to a broad audience.

**Weaknesses:**

1. Leveraging LLMs with simple prompts to generate checklists is a straightforward approach. Previous work has also used decomposition techniques to evaluate responses across multiple dimensions, similar to step-by-step verification of LLMs' instruction-following abilities. While this method has been applied to various evaluation metrics, to my knowledge, this is the first time it has been specifically focused on instruction-following.
2. The construction details and statistics of the Internal dataset are not sufficiently explained, which reduces confidence in the reliability of the results when using LLMs for checklist generation.
3. When evaluating the generated checklists against gold labels, the authors use metrics like ROUGE and BLEU. However, these metrics are less effective in knowledge-intensive contexts, suggesting a need for additional manual annotation or alternative metrics. However, the human annotation results are missed.
4. The preference labeling approach of annotators does not fully align with the checklist-based method for evaluating instruction-following capabilities. Human annotation will consider the quality of the response while TICK only considers instruction-following ability.
5. The low inter-annotator agreement for direct scoring raises concerns, as the authors only demonstrate TICK's effectiveness through pairwise correlation with human annotations. If the inter-annotator agreement for pairwise scoring is similarly low, it might undermine the validity of this correlation.
6. The comparison of TICK to other evaluation methods is limited to direct scoring and an ablated version (Check-then-Score). This restricts the scope of the comparison. Evaluations with fine-tuned models or well-established frameworks could provide a fairer assessment.
7. In self-refinement experiments, the baseline comparison is limited to vanilla self-refinement, which is insufficient. Incorporating additional strong baselines would provide a more comprehensive understanding of STICK's effectiveness.

Reference:

Kalpesh Krishna, Aurko Roy, and Mohit Iyyer. Hurdles to progress in long-form question answering, 2021. URL https://arxiv.org/abs/2103.06332.

Shashank Sonkar and Kangqi Ni and Lesa Tran Lu and Kristi Kincaid and John S. Hutchinson and Richard G. Baraniuk. Automated Long Answer Grading with RiceChem Dataset, 2024. URL https://arxiv.org/abs/2404.14316

**Questions:**

1. The caption for Figure 3(a) appears to be out of sequence or unclear. Could the authors clarify or reorder the content for better coherence?
2. The self-refinement process using STICK results in a minor decline in the last iteration, could the authors make a further explanation?

**Details Of Ethics Concerns:**

There is a potential risk that using STICK for harmful instructions (e.g., those involving discrimination or violence) may increase the harmfulness of LLM responses. Ethical safeguards should be considered to mitigate such issues.

---

> ### Author Response · Authors · 2024-11-13
> **Addressing concerns**
>
> We would like to thank the reviewer for their clear and focused review. We are glad that the reviewer sees the presented method as “novel and significant”, including “comprehensive experiments”. In light of this, we are surprised that the reviewer’s concerns should warrant the current score and thoroughly address each one below.
>
> > Previous work has also used decomposition techniques...
>
> Whilst it is true that prior work decomposes the evaluation task, our work is the first to take an approach to decomposition that has proven powerful in fixed datasets and fully automate the decomposition and evaluation itself using an LLM. Our work is also the first to show that such a decomposition technique enables in-context self-improvement/ self-correction in settings where unstructured self-critiques fail.
>
> > The construction details and statistics of the Internal dataset are not sufficiently explained.
>
> We thank the reviewer for drawing attention to this and have included further details on both the annotator pool and Internal dataset construction in Appendix H of the updated manuscript. The Internal dataset and its full construction details are scheduled for public release within the next month (footnote 1 of the manuscript).
>
> > The authors use metrics like ROUGE and BLEU.
>
> Due to the expense of human annotation, we were unable to additionally acquire human annotation results for this comparison. As suggested by reviewers KpU7 and sSrg, we have included semantic similarity (BERTScore) between checklists in Table 2 of the updated manuscript, where we see that LLM-generated checklists maintain strong similarity to gold standard human-written checklists.
>
> > The preference labelling approach of annotators does not fully align with the checklist-based method.
>
> We would like to raise two key points that we are confident address this claim. Firstly, as can be seen in the prompt for checklist generation and as is stated in line 174 of the manuscript, the LLM is prompted to include “implicit criteria that are generally important for an instruction’s problem domain” in the checklist. Secondly, the superior agreement with gathered human preferences achieved by TICK relative to asking an LLM-as-judge for a preference or score empirically demonstrates that it is better aligned with the preference labelling approach of annotators.
>
> > The low inter-annotator agreement for direct scoring raises concerns.
>
> TICK’s effectiveness was demonstrated by comparing to preference annotations on Internal, for which we provide the inter-annotator agreement in Appendix H of the updated manuscript (0.684 Krippendorff’s alpha). This difference reflects the fact that WildBench involves particularly long and sometimes low quality instructions, direct scoring yields lower agreement than preference labelling, and annotators are familiar with the Internal instruction set.
>
> > Evaluations with fine-tuned models or well-established frameworks could provide a fairer assessment.
>
> Given that TICK requires no additional data or fine-tuning, we firmly disagree that comparing to fine-tuned evaluator models would be a fairer assessment. As an alternative inference scaled baseline, we have additionally provided results for a majority vote (Maj@K) [1] version of preference evaluation and direct scoring among K=32 parallel samples in Table 4 of the updated manuscript. Notably, both remain inferior to TICK.
>
> > The baseline comparison is limited to vanilla self-refinement, which is insufficient.
>
> Self-refine [2] is itself a relatively new method, with no well-established, fine-tuning free alternatives. There are numerous papers indicating that purely in-context self-refinement in fact generally fails, with a prominent recent paper [3] claiming that RL fine-tuning is absolutely necessary to achieve this behaviour in self-correction settings. Yet, in Table 1 we show that STICK is able to reliably self-correct across almost all task categories in the challenging benchmark LiveBench. We believe that this is a very significant result.
>
> ## Answering questions
>
> 1. We thank the reviewer for identifying a potentially out-of-sequence figure caption, but are unable to identify which they mean. Could the reviewer please clarify whether they mean Table 3 (a) or Figure 3 (which has no subfigure labelled (a))?
>
> 2. As shown in [3, 4] and in Table 1, in-context self-refinement is typically prone to response quality degradation, as the LLM can misidentify issues with its own response. The small performance dip in the fourth iteration on WildBench simply shows that the number of iterations STICK can sustain improvements is still limited.
>
> [1] Wang et al, Self-Consistency Improves Chain of Thought Reasoning in Language Models, 2022
>
> [2] Madaan et al, Self-Refine: Iterative Refinement with Self-Feedback, 2023
>
> [3] Kumar et al, Training Language Models to Self-Correct via Reinforcement Learning, 2024
>
> [4] Huang et al, Large Language Models Cannot Self-Correct Reasoning Yet, 2023

---

> > ### Comment · Reviewer_saAj · 2024-11-24
> >
> > Thank you for your reply, I have decided to raise the score to 5. A higher score still requires additional experiments to address related concerns.

---

> > > ### Author Response · Authors · 2024-11-25
> > > **Thank-you for engagement**
> > >
> > > Thank-you again for engaging with our work and our responses during this discussion period, as well as for increasing your score. We are confident that the additional results have strengthened the paper. We humbly ask if the reviewer could please let us know what particular further experiments they have in mind.
> > >
> > > We believe that the TICK results are especially compelling in light of the new self-consistency baseline, and see demonstrating robust self-correction with STICK as a surprising result in light of claims that purely in-context self-correction does not work in the papers cited in our previous response. Moreover, STICK is shown to be more effective for best-of-N self-selection than a strong external reward model (ArmoRM).

---

### Author Response · Authors · 2024-11-13
**Updates to the manuscript and individual comments**

We would like to thank all reviewers for their insights on the paper and suggestions for improvement. We have taken on all actionable suggestions and updated the manuscript accordingly. We have also submitted individual comments to each reviewer with further details relevant to their specific reviews.

We thank each reviewer in advance for taking the time to read our comments engage in further discussion.

---

### Author Response · Authors · 2024-11-20
**Hoping to engage in discussion**

Thanks again to all the reviewers for your time and effort during the review process. We appreciate that you found our work insightful, and we’re glad that there is excitement about our progress on LLM-as-judge evaluation and using this to enable self-improvement and self-correction in settings where other methods fail. Your thoughtful reviews have helped us dramatically improve the clarity and rigour of our submission.

We have responded to each reviewer individually, and updated the manuscript with new results and points of clarification. If you find our answers responsive to your concerns, we would be grateful if you considered increasing your score, and if you have additional questions, we’re happy to engage further.

We kindly ask that the reviewers respond to our comments during this discussion period.

---

### Author Response · Authors · 2024-11-22
**Request for engagement**

Dear reviewers,

We would like to kindly request that you engage with our responses, as the discussion period nears closing. We are keen to take the opportunity to ensure that each concern is addressed and to continue to improve the paper. Our initial responses were posted 9 days ago and we have thus far only received engagement from a single reviewer, who we thank again for doing so.

Kind regards,
The Authors

---

### Meta-Review · Area_Chair_S7p1 · 2024-12-21

**Metareview:**

The authors propose TICK, a method that leverages LLMs to decompose instructions into checklists of yes/no questions. This idea aligns well with assessment practices. The paper conducts comprehensive automated and manual consistency experiments to evaluate and highlight the advantages of the TICK evaluation method. However, the approach of using LLMs to generate checklists is not novel, and the evaluations based on the checklist offer limited insights.

**Additional Comments On Reviewer Discussion:**

There is a hot discussion with some key issues highlighted:

1. Experimental Setting: The experimental setup, particularly regarding the benchmark, rather than fine-tuned models or evaluation metrics, is well-clarified.
2. Human Annotations: Additional details on the human annotations should be provided.
3. Limited Technical Contribution: In my opinion, the method is quite simple, offering few insights, and the authors have not effectively presented their main contribution.

---

### Decision · Program_Chairs · 2025-01-22

Reject